# RTN3 inhibits RIG-I-mediated antiviral responses by impairing TRIM25-mediated K63-linked polyubiquitination

Ziwei Yang[1,2†], Jun Wang[3†], Bailin He[3†], Xiaolin Zhang[1], Xiaojuan Li[1,2]*, Ersheng Kuang[1,2]*

[1]Institute of Human Virology, Zhongshan School of Medicine, Sun Yat-Sen University, Guangzhou, China; [2]Key Laboratory of Tropical Disease Control (Sun Yat-Sen University), Ministry of Education, Guangzhou, China; [3]Zhongshan School of Medicine, Sun Yat-Sen University, Guangzhou, China

**Abstract** Upon viral RNA recognition, the RIG-I signalosome continuously generates IFNs and cytokines, leading to neutrophil recruitment and inflammation. Thus, attenuation of excessive immune and inflammatory responses is crucial to restore immune homeostasis and prevent unwarranted damage, yet few resolving mediators have been identified. In the present study, we demonstrated that RTN3 is strongly upregulated during RNA viral infection and acts as an inflammation-resolving regulator. Increased RTN3 aggregates on the endoplasmic reticulum and interacts with both TRIM25 and RIG-I, subsequently impairing K63-linked polyubiquitination and resulting in both IRF3 and NF-κB inhibition. Rtn3 overexpression in mice causes an obvious inflammation resolving phenomenon when challenged with VSV, Rtn3-overexpressing mice display significantly decreased neutrophil numbers and inflammatory cell infiltration, which is accompanied by reduced tissue edema in the liver and thinner alveolar interstitium. Taken together, our findings identify RTN3 as a conserved negative regulator of immune and inflammatory responses and provide insights into the negative feedback that maintains immune and inflammatory homeostasis.

*For correspondence:
lixjuan3@mail.sysu.edu.cn (XL);
kuangersh@mail.sysu.edu.cn (EK)

†These authors contributed equally to this work

Competing interests: The authors declare that no competing interests exist.

## Introduction

The innate immune responses can be triggered by pathogen-associated molecular patterns (PAMPs) and then serve as the first line of defense against invading pathogens (*Takeuchi and Akira, 2010*). Upon viral infection, pattern recognition receptors (PRRs) detect viral RNA, DNA and other viral products and subsequently mediate the activation of downstream signaling pathways (*Roers et al., 2016*; *Loo and Gale, 2011*). Both RIG-I and MDA5, two crucial members of the RIG-I-like receptor (RLR) family, sense cytoplasmic viral RNA and activate antiviral responses. Both RIG-I and MDA5 share the same domain architecture, comprising two N-terminal caspase activation recruitment domains (CARDs), a central DEAD box helicase/ATPase domain and a C-terminal regulatory domain (CTD) (*Loo and Gale, 2011*). MDA5 has been reported to recognize longer dsRNA molecules (>2 kb), while RIG-I prefers shorter (< 1–2 kb) or 5′ triphosphate-containing dsRNA (*Peisley et al., 2012*; *Peisley et al., 2011*). Unlike MDA5, which is conformationally unaltered, upon recognition of dsRNA by the CTD domain, RIG-I unfolds and releases its CARDs from the inward folding state formed by its helicase domain, thereby transforming into an activated state (*Jiang et al., 2011*; *Kowalinski et al., 2011*). The CARD domain then recruits MAVS and activates the downstream signaling cascade to induce the expression of type I interferons, IFN stimulated genes (ISGs), inflammatory cytokines, and/or chemokines (*Kowalinski et al., 2011*).

Considering that RIG-I-mediated innate antiviral responses are important, drastic and rapid upon viral RNA recognition, multiple mechanisms have evolved to precisely regulate RIG-I-mediated

antiviral signaling to maintain the balance between immunity and tolerance and prevent severe or even fatal unnecessary damage, which may be caused by excessive immune and inflammatory responses (*Eisenächer and Krug, 2012*). Some modifications of RIG-I reduce its stability. For example, LRRC25 targets RIG-I for ISG15-associated autophagy degradation (*Du et al., 2018*), whereas the E3 ubiquitin ligases RNF125, STUB1, c-Cbl, and CHIP induce the K48-linked polyubiquitination of RIG-I to promote its proteasome-dependent degradation (*Arimoto et al., 2007*; *Zhao et al., 2016*). Alternatively, the posttranslational modification of RIG-I upon K63-linked polyubiquitination has been well shown to regulate RIG-I activation. TRIM25 and Riplet induce this modification in RIG-I (*Gack et al., 2007*; *Oshiumi et al., 2010*), whereas the deubiquitination enzyme USP3 targets the RIG-I CARDs to cleave its K63-linked polyubiquitin (*Cui et al., 2014*). In addition, CKII and PKCα/β phosphorylate the RIG-I CTD and CARDs to negatively regulate its activation (*Sun et al., 2011*; *Maharaj et al., 2012*).

Although the acute immune response and inflammation defend against viral infection and eliminate virus-induced damage, excessive and unrestricted responses can lead to excessive tissue injury and even organ failure. Several diseases have been shown to be related to chronic inflammation or autoimmunity, such as asthma, arthritis, periodontal disease, and neurodegenerative disorders (*Chiang et al., 2017*; *Chiang and Serhan, 2017*; *Sugimoto et al., 2016*; *Alessandri et al., 2013*). Inflammation resolution has been shown to be an essential active process (*Buckley et al., 2013*), and pro-resolving mediators regulate inflammation resolution through multiple strategies, including the inhibition of signal transduction, the regulation of cytokines and chemokines, the cessation of leukocyte infiltration and the clearance of apoptotic cells (*Sugimoto et al., 2016*; *Ortega-Gómez et al., 2013*).

Viral infections also trigger different ER stresses due to the excessive synthesis and accumulation of viral proteins in the ER lumen or through signal hijacking (*Sano and Reed, 2013*). In response to ER stress, two transcription factors, CHOP and ATF6, are activated, and the translational regulator eIF2α is phosphorylated and subsequently reprograms cellular transcription and translation (*Oyadomari and Mori, 2004*; *Walter et al., 2011*). Although reticulon family members have shown to comprise a highly conserved housekeeping protein family in multiple species (*Moreira et al., 1999*), one reticulon member, RTN3, is transcriptionally upregulated under ER stress by CHOP or ATF6 (*Wan et al., 2007*).

In our previous unpublished experiments, we observed that RTN3 is dramatically upregulated upon RNA viral infection when using VSV as a stimulus. However, the mechanism and function of RTN3 upregulation during viral infection is unclear, raising the strong possibility that RTN3 is involved in antiviral and inflammatory responses. In the present study, we showed that RTN3 acts as a conserved negative regulator and suppresses RIG-I-mediated immune and inflammatory responses during RNA viral infections. We reveal that RTN3 exerts an inhibitory effect on RIG-I signalosome activation by impairing TRIM25-mediated RIG-I K63-linked polyubiquitination. Our study may contribute to a better understanding of a novel mechanism for inflammation resolution and the reconstitution of tissue homeostasis during viral infections.

## Results

### RTN3 is upregulated and self-aggregates upon RNA viral infection

In our previous study, we occasionally observed increased RTN3 expression upon vesicular stomatitis virus (VSV) infection, leading us to investigate why RNA viral infection increases RTN3 expression. Wild-type (WT) HEK293T cells were infected with enhanced GFP-tagged vesicular stomatitis virus (VSV-eGFP) or treated with the dsRNA synthetic analog poly(I:C) at different time points. We observed that RTN3 protein levels were dramatically increased during VSV-eGFP infection (*Figure 1A*, top) or under poly(I:C) stimulation (*Figure 1A*, bottom), with both conditions showing the same pattern of *RTN3* mRNA upregulation (*Figure 1B and C*, top) accompanied by increased *TNF* mRNA levels that were used to indicate the effectiveness of VSV-eGFP or poly(I:C) treatment toward inflammatory induction in stimulated cells (*Figure 1B and C*, bottom). Interestingly, we observed that both TNF-α and IFN-β could upregulate RTN3 expression at both the protein (*Figure 1D*) and mRNA levels (*Figure 1E*). In addition, human PBMCs from healthy donors infected

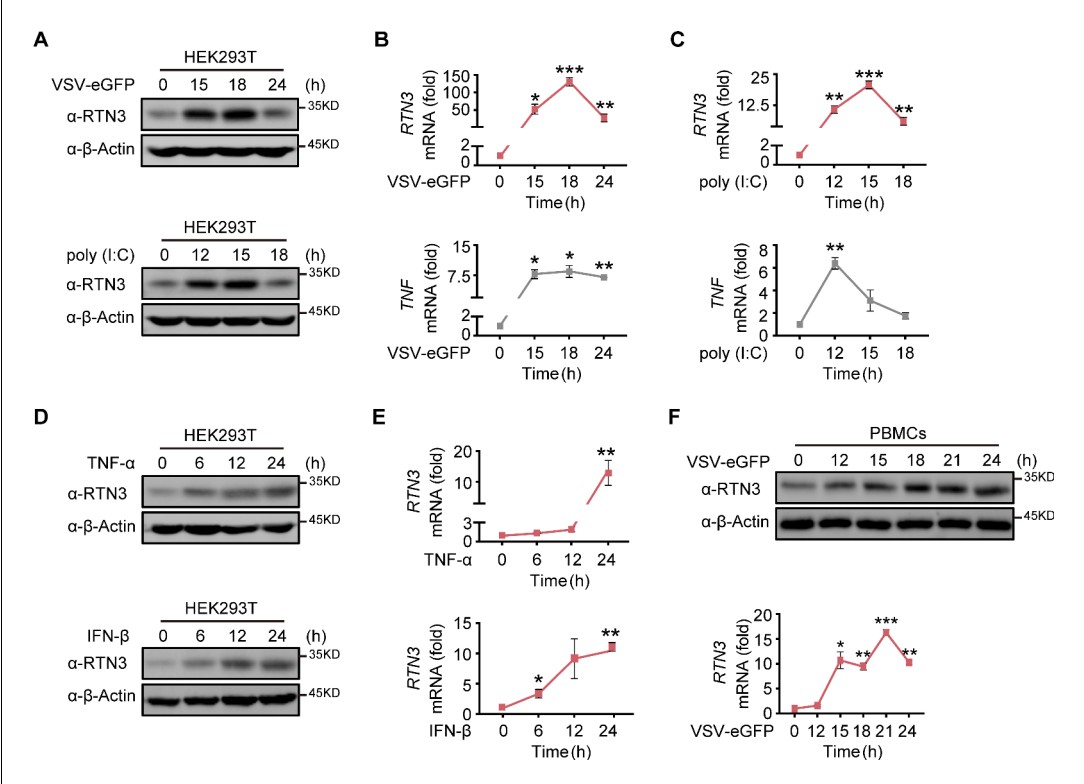

**Figure 1.** RTN3 is upregulated and self-aggregates upon RNA viral infection. (A) Immunoblot analysis of HEK293T cells infected with VSV-eGFP (MOI = 1) or treated with poly(I:C) (5 µg/ml) at the indicated timepoints. (B–C) mRNA levels of *RTN3* and *TNF* in the same samples shown in (A) were detected by real-time PCR. B. VSV-eGFP-infected group, C. poly(I:C)-treated group. (D) Immunoblot analysis of HEK293T cells treated with TNF-α (10 ng/ml) or IFN-β (20 ng/ml) at the indicated timepoints. (E) mRNA levels of *RTN3* in the same samples shown in (D) were detected by RT-PCR. (F) Immunoblot analysis of PBMCs infected with VSV-eGFP (MOI = 2) at the indicated timepoints (top), and mRNA levels of *RTN3* in the same samples were detected by RT-PCR (bottom). In (A, D, F), the data are representative of three independent experiments. In (B, C, E, F), the data are shown as the mean values ± SD (n = 3). *, p < 0.0332; **, p < 0.0021; ***, p < 0.0002; and ****, p < 0.0001 by Sidak's multiple comparisons test.

The online version of this article includes the following source data and figure supplement(s) for figure 1:

**Source data 1.** Uncropped blots with bands labeled AI file and raw unedited blots files for panels A, D, and F.

**Source data 2.** Excel file of RT-PCR measurements and description of statistical tests for panel B.

**Source data 3.** Excel file of RT-PCR measurements and description of statistical tests for panel C.

**Source data 4.** Excel file of RT-PCR measurements and description of statistical tests for panel E.

**Source data 5.** Excel file of RT-PCR measurements and description of statistical tests for panel F.

**Figure supplement 1.** RTN3 is upregulated and self-aggregates upon RNA viral infection.

**Figure supplement 1—source data 1.** Uncropped blots with bands labeled AI file and raw unedited blots files for panels A, B, D, F, and H.

**Figure supplement 1—source data 2.** Excel file of RT-PCR measurements and description of statistical tests for panel C.

**Figure supplement 1—source data 3.** Excel file of RT-PCR measurements and description of statistical tests for panel E.

**Figure supplement 1—source data 4.** Excel file of images quantification and description of statistical tests for panel G.

with VSV exhibited an accordant upregulation of RTN3 expression (*Figure 1F*). These results suggest that RTN3 levels are increased during RNA viral infection.

Since RTN3 is conserved and ubiquitously expressed in various tissues, we further assessed the pattern of RTN3 upregulation in multiple cell lines, including A549 and THP-1 cells. As expected, the protein (*Figure 1—figure supplement 1A and B*) and mRNA (*Figure 1—figure supplement 1C*) levels of RTN3 were both significantly increased by poly(I:C) stimulation or VSV-eGFP infection in A549 cells, with a similar phenomenon observed in THP-1 cells (*Figure 1—figure supplement 1D and E*). The upregulation of RTN3 expression was greatly attenuated in CHOP knockout cells compared with wild-type cells under VSV infection (*Figure 1—figure supplement 1F*). Notably, confocal microscopy analysis of HeLa cells transfected with mCherry-tagged RTN3 followed by challenge with poly(I:C) showed that RTN3 was localized on the endoplasmic reticulum (ER), while under poly(I:C)

stimulation, RTN3 proteins converged and formed many aggregated bodies of varying sizes (*Figure 1—figure supplement 1G*). The coimmunoprecipitation and immunoblot analysis with GFP-tagged RTN3 and HA-tagged RTN3 further proved the existence of this aggregated body (*Figure 1—figure supplement 1H*). It was further confirmed that RTN3 was colocalized with calnexin on the ER as well as their tight colocalization within the aggregation under poly(I:C) stimulation, which substantiated that RTN3 aggregation was localized on the endoplasmic reticulum (ER) (*Figure 1—figure supplement 1I*). Taken together, these data suggest that RTN3 expression is induced by RNA viral infection and its observed upregulation and self-aggregation indicate that RTN3 may be involved in regulating innate immune and inflammatory responses.

## RTN3 overexpression suppresses antiviral immune responses

To investigate the function of RTN3 in regulating innate immune responses, we performed luciferase assays using ISRE-luc, IFNβ-Luc, and NF-κB-Luc reporters and observed that RTN3 markedly inhibited all activities induced by RIG-I overexpression (*Figure 2A*). In addition, poly(I:C)- or Sendai virus (SeV)-stimulated ISRE-luc activities were both restrained by RTN3 overexpression in a dose-dependent manner (*Figure 2B*). However, RTN3 overexpression had a slightly negative effect on MDA5-induced ISRE-luc activities (*Figure 2—figure supplement 1A*) and barely inhibited TLR3-induced ISRE-luc activities (*Figure 2—figure supplement 1B*). Constant cGAS-STING or IRF3-induced ISRE-luc activities were observed upon RTN3 overexpression, excluding the possibility that RTN3 inhibits antiviral responses by targeting cGAS-STING-IRF3-axis (*Liu et al., 2012*; *Figure 2—figure supplement 1C and D*). These data reveal that RTN3 primarily suppresses RIG-I-mediated antiviral immune responses.

To further demonstrate the attenuation of antiviral activities by RTN3, HEK293T cells were transfected with an RTN3-encoding plasmid and subsequently infected with VSV-eGFP. Fluorescence microscopy and flow cytometry results showed that RTN3 overexpression rendered the cells highly susceptible to viral infection and augmented virus amplification compared to the empty vector-transfected cells, which displayed a less sensitive phenotype (*Figure 2C and D*), and the MTS assay performed with control cells and RTN3-overexpressing HEK293T cells for over 36 hr showed that the cell viability was not affected by RTN3 overexpression (*Figure 2—figure supplement 1E*). A consistent result was also observed in long-term puromycin-selected, stable HA-tagged RTN3 vs. HA-tagged empty vector-expressing THP-1 cells (THP-1^HA-RTN3 vs. THP-1^HA-Ev) (*Figure 2—figure supplement 1F*), which excluded the possibility that ectopic RTN3 expression may transiently overload the endoplasmic reticulum (ER) and impair the translation of antiviral proteins and cell survival.

To further assess the role of RTN3 in attenuating antiviral activities, we transfected HEK293T cells with RTN3 or empty vector and followed by infection with VSV-eGFP. The phosphorylation of IKKα/β, TBK1, p65, and IRF3 was dramatically decreased in RTN3-overexpressing cells compared to that observed in the control groups (*Figure 2E*, left). Quantitative analysis by gray intensity scanning of immunoblot bands further demonstrated the significant decreases in the phosphorylation of the two kinases and two transcription factors (*Figure 2E*, right).

Accordingly, we investigated whether RTN3 suppresses type I IFN- and interferon-stimulated gene (ISG) expression. RT-PCR results showed that RTN3-overexpressing cells exhibited a deficiency in *IFNB1*, *IFIT2*, *IFIT1*, *TNF* and the proinflammatory cytokines *CCL20* and *IL6* during VSV infection (*Figure 2F*). Similarly, the mRNA levels of *IFNB1, IFIT2*, and *IFIT1* were downregulated to a great extent in THP-1^HA-RTN3 cells compared to THP-1^HA-Ev cells after VSV infection (*Figure 2—figure supplement 1G*). Because the amounts of the proinflammatory cytokines *CCL20* and *IL6* were both markedly decreased by RTN3 overexpression under dsRNA stimulation [VSV or poly(I:C)], we assessed whether any other inflammatory cytokines or related genes were altered by RTN3 overexpression. To this end, we performed RT-PCR array analysis and observed that many inflammatory factors and genes were downregulated during RTN3 overexpression (*Figure 2—figure supplement 1H*). These results suggest that RTN3 overexpression preferentially suppresses RIG-I-mediated antiviral immune and inflammatory responses.

## RTN3 depletion augments antiviral immune responses

To further elucidate the physiological function of RTN3 in antiviral responses, we depleted its expression by short hairpin RNAs (shRNAs) in human healthy donor PBMCs (*Figure 3A*), in which

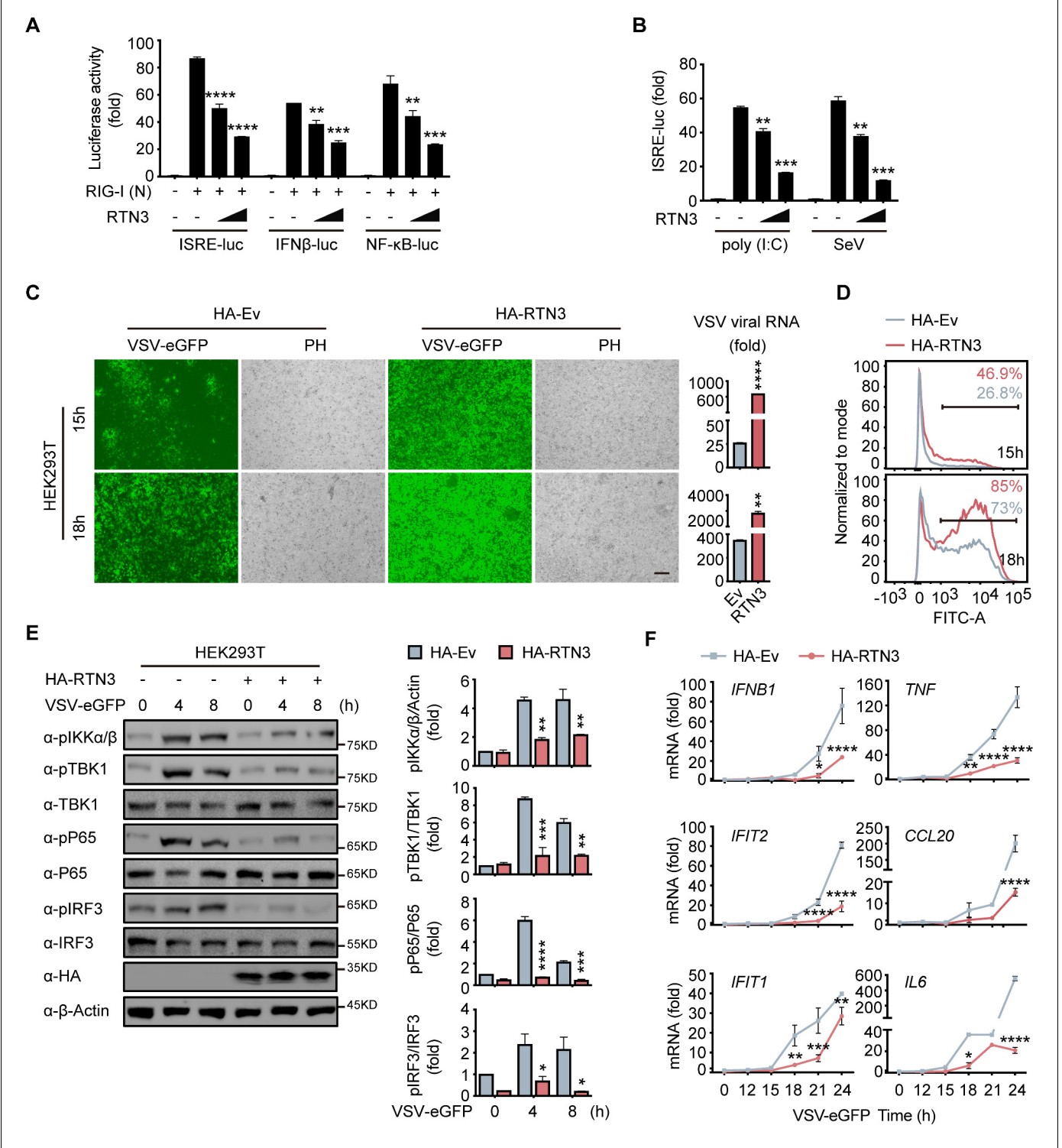

**Figure 2.** RTN3 overexpression suppresses antiviral immune responses. (**A**) Luciferase assays of HEK293T cells transfected with an ISRE luciferase reporter (ISRE-Luc), an IFNB luciferase reporter (IFNβ-Luc) or an NF-κB luciferase reporter (NF-κB-Luc) together with an HA-tagged empty vector (HA-Ev, no wedge) or increasing amounts (wedge) of plasmid encoding RTN3 (HA-RTN3) followed by transfection with a plasmid encoding the RIG-I CARD domain [RIG-I (N)] to activate the pathway. Cell lysates were collected 24 hr posttransfection, the same as that described in the following experiments, if no additional annotation was performed. (**B**) Luciferase activity of HEK293T cells transfected with ISRE-Luc, together with HA-Ev or increasing amounts of plasmid HA-RTN3, followed by treatment with poly(I:C) (5 µg/ml) or SeV (MOI = 0.1) for 16 or 24 hr. (**C**) Fluorescence and phase contrast (PH) analyses of HEK293T cells transfected with HA-Ev or HA-RTN3, followed by infection with VSV-eGFP (MOI = 0.05) at the indicated timepoints (left).

*Figure 2 continued on next page*

*Figure 2 continued*

Scale bar, 100 μm. mRNA levels of VSV viral RNA in the same samples were analyzed by RT-PCR (right). (D) The percentage of eGFP-positive cells in the same samples shown in (C) was analyzed by flow cytometry. (E) Immunoblot analysis of HEK293T cells transfected with HA-Ev or HA-RTN3 followed by infection with VSV-eGFP (MOI = 1) at the indicated timepoints (left). Quantitative comparison of the indicated protein levels analyzed by gray intensity scanning of blots (right). (F) RT-PCR analysis of *IFNB1, IFIT2, IFIT1, TNF, IL6,* and *CCL20* mRNA levels in HEK293T cells transfected with HA-Ev or HA-RTN3 followed by infection with VSV-eGFP (MOI = 1) at the indicated timepoints. In (E), the data are representative of three independent experiments. In (A, B, C, F), the data are shown as the mean values ± SD (n = 3). *, p < 0.0332; **, p < 0.0021; ***, p < 0.0002; and ****, p < 0.0001 by Sidak's multiple comparisons test.

The online version of this article includes the following source data and figure supplement(s) for figure 2:

**Source data 1.** Uncropped blots with bands labeled AI file and raw unedited blots files for panel E.
**Source data 2.** Excel file of luciferase reporter assay measurements and description of statistical tests for panel A.
**Source data 3.** Excel file of luciferase reporter assay measurements and description of statistical tests for panel B.
**Source data 4.** Excel file of RT-PCR measurements and description of statistical tests for panel C.
**Source data 5.** Excel file of blots quantification and description of statistical tests for panel E.
**Source data 6.** Excel file of RT-PCR measurements and description of statistical tests for panel F.
**Figure supplement 1.** RTN3 suppresses RNA virus-induced antiviral immune and inflammatory responses.
**Figure supplement 1—source data 1.** Excel file of luciferase reporter assay measurements and description of statistical tests for panel A.
**Figure supplement 1—source data 2.** Excel file of luciferase reporter assay measurements and description of statistical tests for panel B.
**Figure supplement 1—source data 3.** Excel file of luciferase reporter assay measurements and description of statistical tests for panel C.
**Figure supplement 1—source data 4.** Excel file of luciferase reporter assay measurements and description of statistical tests for panel D.
**Figure supplement 1—source data 5.** Excel file of MTS assay measurements and description of statistical tests for panel E.
**Figure supplement 1—source data 6.** Excel file of RT-PCR measurements and description of statistical tests for panel F.
**Figure supplement 1—source data 7.** Excel file of RT-PCR measurements and description of statistical tests for panel G.
**Figure supplement 1—source data 8.** Excel file of RT-PCR measurements and description of statistical tests for panel H.
**Figure supplement 1—source data 9.** Excel file of RT-PCR primers list for panel H.

RTN3 normally undergoes upregulation during viral infection (*Figure 1F*). Within our experience, complete or vast obliteration of the expression of RTN3 by knockout or knockdown leads to cell death; hence, RTN3 expression were moderately depleted with transient shRNA transduction (*Figure 3A*). PBMCs were infected with control or *RTN3* shRNA-encoding lentivirus for 12 hr, and the infected cells were then settled for the following 24 hr and subsequently infected with VSV-eGFP. Consistent with the decreased phosphorylation levels of TBK1, p65 and IRF3 in RTN3 overexpressing cells, RTN3 depletion significantly augmented the phosphorylation and activation of these critical kinases and transcription factors upon RNA viral infection (*Figure 3B*). Further RT-PCR analyses determined that the levels of *IFNB1, IFIT2, IFIT1, TNF* and the proinflammatory cytokines *CCL20* and *IL6* were enhanced by *RTN3* knockdown during VSV infection (*Figure 3C*), while the amplification of VSV was compromised (*Figure 3D*). Collectively, depressed RTN3 expression facilitates antiviral innate immune and inflammatory responses upon viral infection.

## RTN3 interacts with TRIM25 or RIG-I

Next, we investigated the potential mechanism for the RTN3-mediated inhibition of antiviral responses. We noted that RTN3 may function as a tripartite motif-containing protein 25 (TRIM25)-binding protein through mass spectrometry analysis (*Choudhury et al., 2017*), and TRIM25 is a crucial E3 ligase for the K63-linked polyubiquitination modification of RIG-I to promote its activation upon RNA viral infection (*Gack et al., 2007*). To elucidate the relationship between RTN3 and TRIM25 or RIG-I, HEK293T cells were transfected with GFP-tagged RTN3 and Flag-tagged TRIM25 followed by stimulation with or without VSV and then subjected to coimmunoprecipation and immunoblot analysis. The results showed that RTN3 interacted with TRIM25 under basal conditions and was slightly strengthened upon viral infection (*Figure 4A*, *Figure 4—figure supplement 1A*). Considering that the RTN3 overexpression leads to its aggregation in the ER and may cause nonspecific interactions (such as structural enfolding), we used GFP-TRIM25 to pull down endogenous RTN3 and confirmed its interaction with TRIM25 and that this interaction was strengthened by VSV infection (*Figure 4—figure supplement 1B*). Since TRIM25 directly targets RIG-I, the coimmunoprecipitation assays were performed in *RIG1* knockout HEK293T cells and the results showed that the TRIM25-RTN3 interaction did not require RIG-I, thus we concluded that RTN3 binds to TRIM25

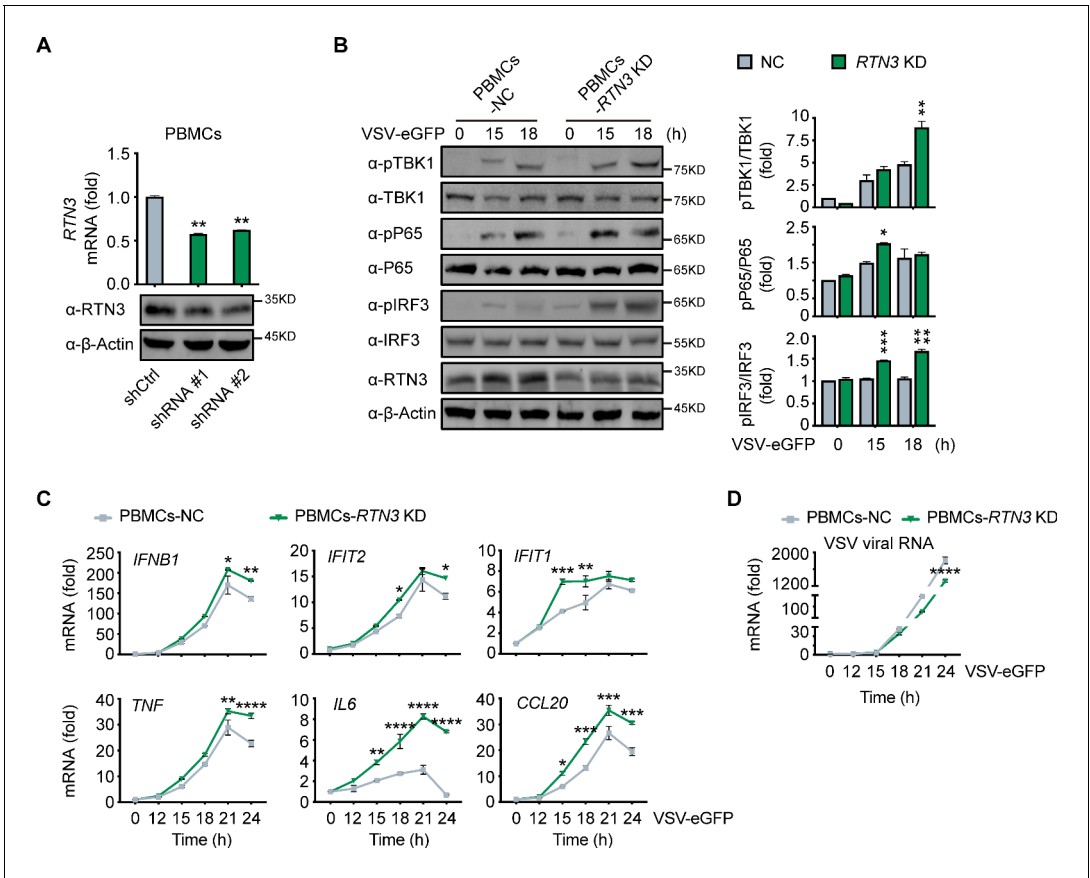

**Figure 3.** RTN3 knockdown enhances antiviral immune responses. (**A**) RT-PCR analysis (top) and immunoblotting analysis (bottom) of *RTN3* mRNA levels in PBMCs infected with Ctrl or *RTN3* shRNA encoding lentivirus (shCtrl, shRNA #1, shRNA #2) for 12 hr and cultured for additional 24 hr (top). (**B**) Immunoblotting analys of PBMCs infected with Ctrl or *RTN3* shRNA encoding lentivirus for 12 hr and cultured for another 24 hr, followed by infection with VSV-eGFP (MOI = 2) at the indicated timepoints (left). Quantitative comparison of the indicated protein levels was analyzed by gray intensity scanning of blots and is shown (right). (**C–D**) RT-PCR analysis of *IFNB1*, *IFIT2*, *IFIT1*, *TNF*, *IL6*, and *CCL20* mRNA levels (**C**) and VSV viral genomic RNA mRNA levels (**D**) in PBMCs infected with Ctrl or *RTN3* shRNA encoding lentivirus for 12 hr and cultured for another 24 hr, followed by infection with VSV-eGFP (MOI = 2) at the indicated timepoints. In (**B**), the data are representative of three independent experiments. In (**A, C, D**), the data are shown as the mean values ± SD (n = 3). *, $p < 0.0332$; **, $p < 0.0021$; ***, $p < 0.0002$; and ****, $p < 0.0001$ by Sidak's multiple comparisons test.

The online version of this article includes the following source data for figure 3:

**Source data 1.** Uncropped blots with bands labeled AI file and raw unedited blots files for panels A and B.

**Source data 2.** Excel file of RT-PCR measurements and description of statistical tests for panel A.

**Source data 3.** Excel file of blots quantification and description of statistical tests for panel B.

**Source data 4.** Excel file of RT-PCR measurements and description of statistical tests for panel C.

**Source data 5.** Excel file of RT-PCR measurements and description of statistical tests for panel D.

independent of RIG-I and excluded the possibility that RTN3 may interact with TRIM25 through the interaction between TRIM25 and RIG-I (*Figure 4B*). Besides, we speculated that RTN3 may also interact with RIG-I. Therefore, we performed coimmunoprecipation assays and observed an interaction between TRIM25 and RIG-I (*Figure 4C*) as well as a similar VSV-promoting pattern as that detected for RTN3 interacting with TRIM25 (*Figure 4—figure supplement 1C*). Furthermore, confocal microscopy results showed the colocalization of RTN3 with RIG-I or TRIM25 on the extended ER framework (*Figure 4D and E*), indicating that RTN3 aggregation may provide a scaffold for TRIM25 oligomerization or the RIG-I signalosome complex. Immunoprecipitation using anti-RTN3 antibody in human PBMCs infected with VSV demonstrated endogenous interaction between RTN3 and TRIM25/RIG-I (*Figure 4F*). Collectively, these results suggested that RTN3 interacts with both RIG-I and TRIM25 and that this interaction could be enhanced by RNA viral infection.

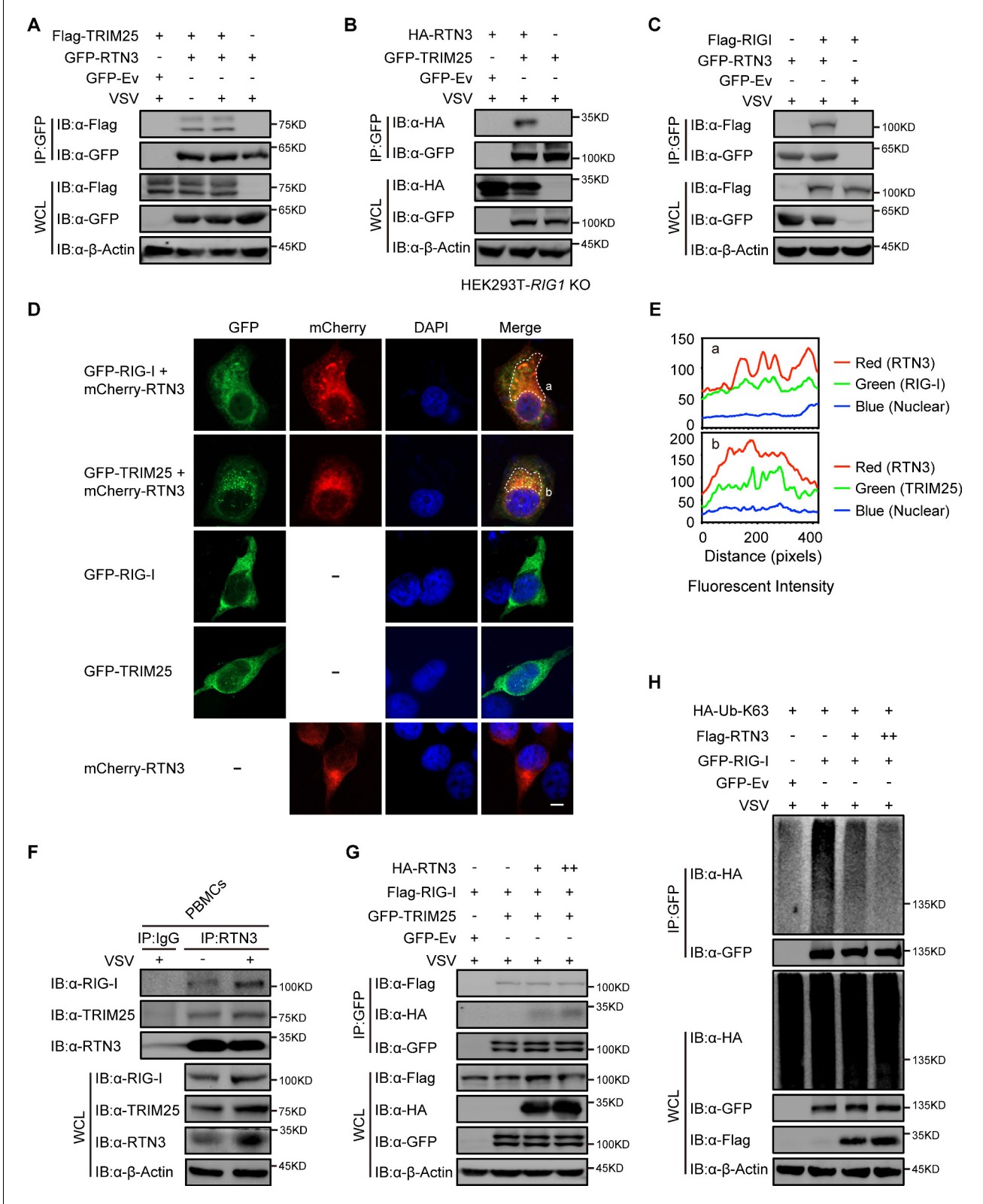

**Figure 4.** RTN3 separately interacts with TRIM25 and RIG-I and inhibits the K63-linked polyubiquitination of RIG-I. (**A**) Coimmunoprecipitation (CoIP) and immunoblot analyses of HEK293T cells transfected with a plasmid encoding TRIM25 (Flag-TRIM25) or a Flag-tagged empty vector (Flag-Ev) together with GFP-tagged RTN3 (GFP-RTN3) or a GFP-tagged empty vector (GFP-Ev) in the indicated groups for 24 hr and followed by infection with VSV (MOI = 1) for 8 hr. (**B**) CoIP and immunoblot analyses of *RIG1* KO HEK293T cells transfected with HA-Ev or HA-RTN3 together with GFP-tagged

*Figure 4 continued on next page*

*Figure 4 continued*

TRIM25 (GFP-TRIM25) or GFP-Ev for the indicated groups for 24 hr and followed by infection with VSV (MOI = 1) for 8 hr. (C) CoIP and immunoblot analyses of HEK293T cells transfected with a plasmid encoding RIG-I (Flag-RIG-I) or Flag-Ev together with GFP-RTN3 or GFP-Ev in the indicated groups for 24 hr and followed by infection with VSV (MOI = 1) for 8 hr. (D) Confocal microcopy analysis of HeLa cells transfected with GFP-RIG-I and mCherry-tagged RTN3 (mCherry-RTN3) or GFP-TRIM25 with mCherry-RTN3, and GFP-RIG-I, GFP-TRIM25, or mCherry-Ev, respectively. The dotted box (a, b) indicates the colocalization area. Scale bar, 5 µm. (E) Qualitative analysis of the fluorescence intensity of the selected area in (D—a, b). (F) Co-IP and immunoblot analyses of PBMCs infected with VSV-eGFP (MOI = 2) for 18 hr, cell lysates were immunoprecipitated with anti-RTN3 or IgG isotype. (G) IP and immunoblot analyses of HEK293T cells transfected with HA-Ev or increasing amounts of HA-RTN3 (+, ++) together with Flag-RIG-I and GFP-tagged TRIM25 (GFP-TRIM25) or GFP-Ev in the indicated groups for 24 hr and followed by infection with VSV (MOI = 1) for 8 hr. (H) CoIP and immunoblot analyses of HEK293T cells transfected with HA-Ub-K63 and Flag-Ev or increasing amounts of Flag-RTN3 (+, ++) together with GFP-RIG-I or GFP-Ev in the indicated groups for 24 hr and followed by infection with VSV (MOI = 1) for 8 hr. In (A–C) and (F–H), the data are representative of three independent experiments.

The online version of this article includes the following source data and figure supplement(s) for figure 4:

**Source data 1.** Uncropped blots with bands labeled AI file and raw unedited blots files for panels A, B, C, F, G, and H.

**Figure supplement 1.** RTN3 has no influence on the protein levels of RIG-I and TRIM25 but affects their K63-polyubiquitination.

**Figure supplement 1—source data 1.** Uncropped blots with bands labeled AI file and raw unedited blots files for panels A, B, C, D, E, F, G, H, and I.

## RTN3 impairs the K63-linked polyubiquitination of RIG-I upon RNA viral infection

We next assessed how RTN3 suppresses RIG-I-mediated antiviral responses through its interaction with TRIM25. Interestingly, the protein levels of RIG-I and TRIM25 were not decreased by RTN3 overexpression in HEK293T cells (*Figure 4—figure supplement 1D*), excluding the possibility that RTN3 may target RIG-I or TRIM25 for their degradation. We also assessed whether the interaction between RIG-I and TRIM25 was inhibited or promoted by RTN3 overexpression. To this end, HEK293T cells were cotransfected with Flag-RIG-I- and GFP-TRIM25-expressing constructs or GFP-tagged empty vector (GFP-Ev) together with increasing amounts of RTN3-expressing plasmid, which was followed by VSV infection. Unfortunately, the interaction between RIG-I and TRIM25 was completely unchanged (*Figure 4G*), indicating that RTN3 negatively regulates RIG-I-mediated antiviral responses via other mechanisms.

Subsequently, we investigated the polyubiquitination of RIG-I upon VSV infection in the absence or presence of RTN3 overexpression. As shown in *Figure 4H* and *Figure 4—figure supplement 1E*, the K63-linked ubiquitination level of RIG-I was greatly decreased by RTN3 overexpression, potentially suppressing RIG-I signalosome formation and downstream signal transduction. In contrast, RTN3 depletion enhanced the K63-linked polyubiquitination level of RIG-I (*Figure 4—figure supplement 1F and G*). We further investigated the ubiquitination of TRIM25 in the presence of RTN3, as the autoubiquitination (K63-linked) of TRIM25 is essential for its catalytic activity (*Gupta et al., 2018*). Following the increasing expression of RTN3 in HEK293T cells, the K63-linked polyubiquitination of TRIM25 was significantly promoted (*Figure 4—figure supplement 1H*), consistent with the observed RTN3-mediated increase in the WT polyubiquitination of TRIM25 (*Figure 4—figure supplement 1I*). Taken together, these data indicate that RTN3 suppresses RIG-I-mediated antiviral innate immune responses by inhibiting K63-linked polyubiquitination upon viral RNA recognition, simultaneously suppressing the K63-linked ubiquitination of RIG-I and promoting the K63-linked polyubiquitination of TRIM25.

## RTN3 inhibition of antiviral responses is TRIM25-dependent

To elucidate whether TRIM25 promotes the ability of RTN3 to inhibit RIG-I-mediated immune responses, we further investigated whether TRIM25 deficiency abolishes the inhibition of RIG-I-mediated antiviral immune responses. To this end, endogenous TRIM25 expression was depleted in HEK293T cells by shRNA (HEK293T-shTRIM25), after which the phosphorylation of TBK1, p65, and IRF3 was detected in both HEK293T-shTRIM25 and HEK293T-shCtrl cells. Compared to the HEK293T-shCtrl cells, TRIM25 depletion counteracted the inhibition of TBK1, p65, and IRF3 phosphorylation induced by RTN3 overexpression when challenged with poly(I:C) (*Figure 5A*). Similarly, RTN3 overexpression in HEK293T-shTRIM25 cells only slightly impaired RIG-I K63-linked polyubiquitination upon VSV infection (*Figure 5B*). Taken together, these data suggest that TRIM25 has a crucial role in the RTN3-mediated suppression exerted of RIG-I activation.

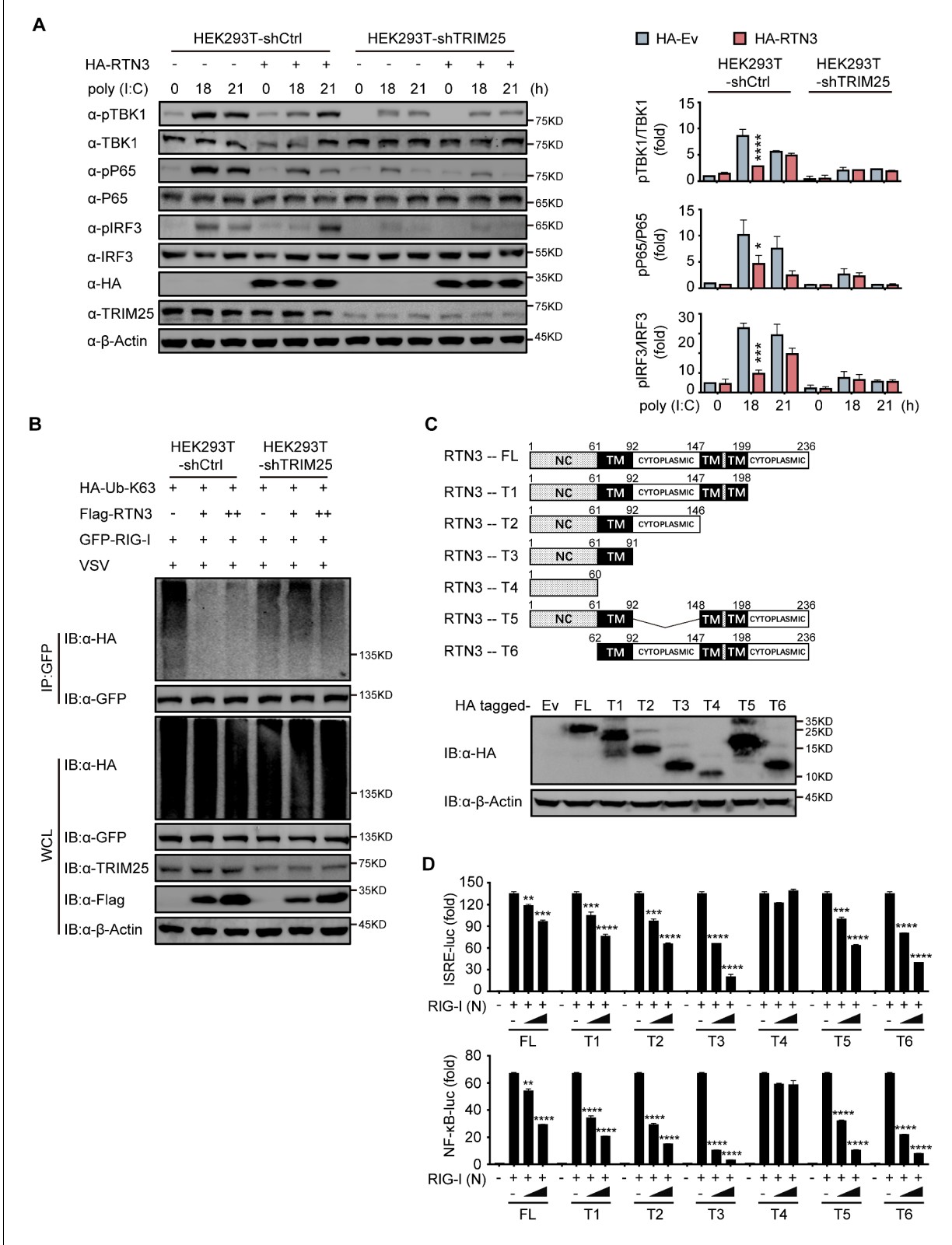

**Figure 5.** RTN3-mediated inhibition of antiviral responses is TRIM25-dependent. (**A**) Immunoblot analysis of control HEK293T cells (HEK293T-shCtrl) and TRIM25 knockdown HEK293T cells (HEK293T-shTRIM25) transfected with HA-Ev or HA-RTN3 and followed by treated with poly(I:C) (5 μg/ml) at the indicated timepoints. (left). Quantitative comparison of the indicated protein levels in (**A**) analyzed by gray intensity scanning of blots (right). (**B**) CoIP and immunoblot analysis of HEK293T-shCtrl and HEK293T-shTRIM25 cells transfected with HA-Ub-K63 and increasing amounts of Flag-RTN3 (+, ++) or

*Figure 5 continued on next page*

*Figure 5 continued*

Flag-Ev together with GFP-RIG-I or GFP-Ev in the indicated groups for 24 hr and followed by infection with VSV (MOI = 1) for 8 hr. (**C**) The structure of RTN3 and its truncated mutants. Noncytoplasmic domain (NC), transmembrane domain (TM), full length (FL), truncated (T) (top). Immunoblot analysis of HEK293T cells transfected with HA-Ev or HA-tagged truncated mutants for 24 hr (bottom). (**D**) Luciferase activity of HEK293T cells transfected with ISRE-Luc (top) and NF-κB-Luc (bottom) together with HA-Ev or increasing amounts (wedge) of HA-RTN3 and its truncated mutants and followed by transfection of RIG-I (N) as an activation of the pathway. In (**A, B**), the data are representative of three independent experiments. In (**D**), the data are shown as the mean values ± SD (n = 3). *, p < 0.0332; **, p < 0.0021; ***, p < 0.0002; and ****, p < 0.0001 by Sidak's multiple comparisons test.

The online version of this article includes the following source data and figure supplement(s) for figure 5:

**Source data 1.** Uncropped blots with bands labeled AI file and raw unedited blots files for panels A, B, and C.
**Source data 2.** Excel file of blots quantification and description of statistical tests for panel A.
**Source data 3.** Excel file of luciferase reporter assay measurements and description of statistical tests for panel D.
**Figure supplement 1.** RTN3 inhibitory activity requires its transmembrane domain 1.
**Figure supplement 1—source data 1.** Uncropped blots with bands labeled AI file and raw unedited blots files for panel A, Band C.

Subsequently, we evaluated which domain of RTN3 is responsible for its inhibitory activity toward RIG-I-mediated antiviral responses. RTN3 localizes to the ER membrane and comprises a noncytoplasmic domain (NC) that inserts into the ER lumen, three transmembrane domains (TM) and two cytoplasmic domains. A series of RTN3 truncations were constructed from the C terminus based on its domain structure (*Figure 5C*) and were then assessed for their ability to suppress RIG-I-mediated immune activation through ISRE-luc and NF-κB-luc reporter assays. The TM1-3-truncated construct (T4 mutant) was unable to inhibit RIG-I (N)-induced ISRE-luc activity, while other truncated mutants maintained similar inhibitory activities as full-length RTN3 (*Figure 5D*, top). The same phenomenon was also observed in NF-κB-luc reporter assays (*Figure 5D*, bottom). These results suggest that the first TM domain is required for the inhibitory activity of RTN3. Furthermore, ubiquitination analysis demonstrated that K63-linked polyubiquitination was impaired by the RTN3 mutants T1, T2 and T3 at levels similar to wild-type RTN3, whereas T4 mutants lacked this activity (*Figure 5—figure supplement 1A*). To further determinate the inhibitory function of the truncated mutants, we performed the coimmunoprecipitation assays and founded that T4 mutants no longer interacts with TRIM25 or RIG-I (*Figure 5—figure supplement 1B and C*). The confocal microscopy results showed that T4 mutants localized in the nuclear while other mutants colocalizing with RIG-I and TRIM25 in the cytoplasm as full length RTN3 did (*Figure 5—figure supplement 1D*). Luciferase assays using ISRE-luc and NF-κB-Luc reporters also showed that RTN3's inhibitory function is insusceptible to the deletion of its noncytoplasmic domain or any cytoplasmic domain (*Figure 5D*). Collectively, our data suggest that RTN3 inhibits RIG-I-mediated innate immune antiviral responses in a TRIM25-dependent manner and that transmembrane domain 1 (aa 61–92) of RTN3 is indispensable for this activity, so long as RTN3 localizes on the endoplasmic reticulum membrane its inhibitory function is retained.

## RTN3 overexpression suppresses antiviral immune responses in mice

We next assessed whether RTN3 is upregulated and inhibits RIG-I-mediated innate immune antiviral responses in vivo during viral infection by treating C57BL/6 mice with PBS, poly(I:C) or VSV-eGFP separately via intravenous (I.V.) injection. Real-time PCR results showed that the mRNA levels of *Rtn3* in liver extracts were significantly higher in the VSV- and poly(I:C)-treated mice than in those of the PBS control mice (*Figure 6A*), and Rtn3 protein levels were also markedly upregulated (*Figure 6B*). These results confirm that RTN3 expression is significantly induced by RNA viral infection in mice, at least in the liver.

To further investigate whether RTN3 upregulation inhibits antiviral responses in vivo, mice were hydrodynamically injected with a plasmid encoding Rtn3 (HA-Rtn3) or the empty vector (HA-Ev) and then inoculated with VSV-eGFP via intraperitoneal (I.P.) injection to infect mice for another 12 hr (*Figure 6C*). At 48 hr, as expected, appropriate levels of plasmid-mediated RTN3 overexpression were detected in the livers of mice (*Figure 6D*). After the mice were sacrificed and then bled, the serum samples were incubated with precoated beads, and flow cytometry results showed that the levels of key inflammatory factors, including Mcp-1, IL-12p70, IFN-γ, and TNF-α were consistently decreased in the Rtn3-overexpressing mice upon viral infection, especially IFN-γ and TNF-α, both of which were markedly downregulated (*Figure 6E*). We next analyzed the immune cell population within the liver and PBMCs through flow cytometry analysis. Cells were grouped as indicated

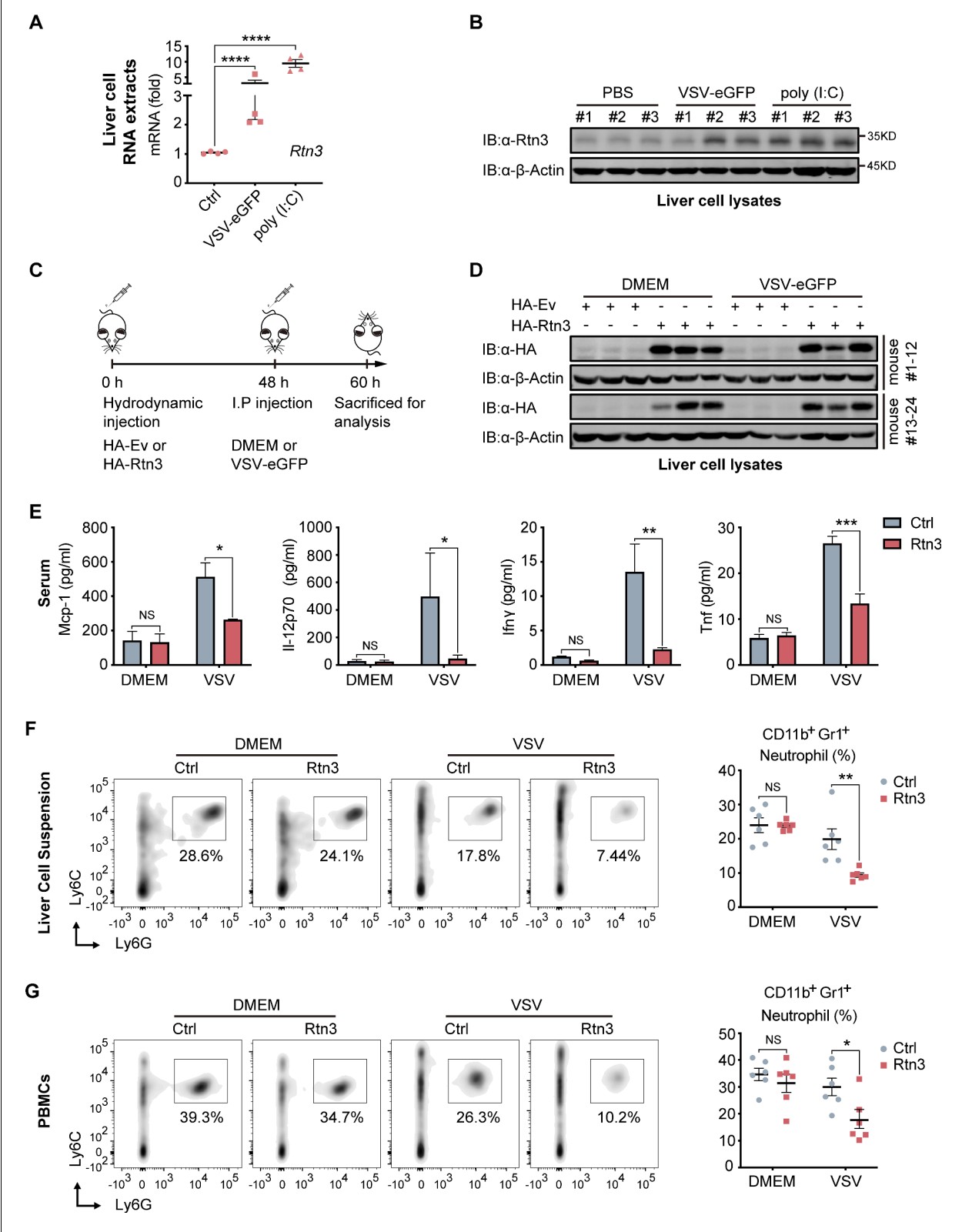

**Figure 6.** RTN3 overexpression suppresses antiviral immune responses in mice. (**A**) mRNA levels of *Rtn3* were detected by RT-PCR in liver cells from mice treated with PBS, poly(I:C) (400 μg/mouse) or VSV-eGFP ($1\times10^7$ PFU/g/mouse) for 12 hr. (**B**) Immunoblot analysis of liver cell lysates of the same samples shown in (**A**), with '# number' indicating individual mice. (**C**) Timeline of in vivo experiments. For plasmid hydrodynamic injection, 20 μg/mouse; for viral infection (intraperitoneal injection), $1\times10^7$ PFU g$^{-1}$/mouse VSV (VSV-eGFP) or 600 μl DMEM /mouse. (**D**) Immunoblot analysis of liver cell lysates

*Figure 6 continued on next page*

*Figure 6 continued*

from the same samples shown in (C). (E) Protein levels of Mcp-1, IL-12p70, Ifn-γ, and Tnf-α in serum samples from the mice shown in (C) were measured by flow cytometry using a BD CBA Mouse Inflammation kit. (F) The percentage of neutrophils in liver cell suspensions from the same mice shown in (C) was detected by flow cytometry. The density graph indicates the cell population groups for each mouse (left), and the percentage comparison for all mice is shown in the scatterplot (right). (G) The percentage of neutrophils among PBMCs from the same mice as in (C) was detected by flow cytometry. The density graph indicates the cell population grouped of each mouse (left), and the percentage comparison for all mice is shown in the scatterplot (right). In (A, E, F, G), the data are shown as the mean values ± SD (n = four in A, n = six in E-G). *, p < 0.0332; **, p < 0.0021; ***, p < 0.0002; and ****, p < 0.0001 by Sidak's multiple comparisons test.

The online version of this article includes the following source data and figure supplement(s) for figure 6:

**Source data 1.** Uncropped blots with bands labeled AI file and raw unedited blots files for panels B and D.

**Source data 2.** Excel file of CBA Mouse Inflammation Kit measurements and description of statistical tests for panel E.

**Figure supplement 1.** Determination of hydrodynamic injection efficiency and flow cytometry analysis of lymphocytes.

(*Figure 6—figure supplement 1A and B*), and upon VSV-eGFP infection, the population of CD11b$^+$-Gr1$^+$ neutrophils (*Figure 6F and G*) dramatically decreased in the Rtn3-overexpressing mice, while other immune cell populations, including CD11$^+$Ly6C$^{hi}$F480$^+$ macrophages, Ly6C$^{hi}$F4/80$^{lo}$ monocytes, CD45$^+$CD3$^+$ T cells and CD45$^+$CD3$^-$CD11b$^+$CD11c$^+$ dendritic cells, showed no significant change (*Figure 6—figure supplement 1C*). Taken together, these results indicate that upon RNA viral infection, RTN3 is upregulated and subsequently suppresses antiviral immune and inflammatory responses, decreasing the population of neutrophils in mice.

## RTN3 overexpression suppresses inflammation in mice

Based on the observed inhibitory activity of RTN3 in mouse experiments, we speculated that RTN3 may promote the resolution of inflammation in vivo. To assess this possibility, the liver and lung tissue sections of mice were stained with hematoxylin and eosin (H and E), and microcopy analysis showed that upon VSV infection, RTN3 overexpression (indicated by RTN3 IHC) alleviated inflammatory cell infiltration and tissue edema in the liver (*Figure 7A* top and middle, 7B, 7C), which was accompanied by a thinning of the alveolar interstitium in the lung (*Figure 7A* bottom, 7D). These results indicated that RTN3 upregulation is induced by RNA viral infection to attenuate inflammation and protect against tissue injury caused by excessive inflammatory responses. Further experiments evaluating VSV-eGFP infection in the liver revealed that the VSV-infected cell population was increased in RTN3-overexpressing mice (*Figure 7—figure supplement 1A and B*), and the MFI of the GFP signal indicated that VSV-eGFP infection also increased to a certain extent. Collectively, these results indicate that RTN3 upregulation under RNA viral infection attenuates tissue inflammation and likely facilitates the resolution of inflammation.

## Discussion

In the present study, with respect to the conserved and ubiquitous expression and ER subcellular localization of reticulon members, we demonstrate that RTN3 is involved in the negative regulation of antiviral responses upon RNA viral infection. Our results demonstrate that RTN3 is upregulated upon infection with VSV or poly(I:C) stimulation and then self-aggregates along the ER membrane. A potential mechanism for RTN3 induction may be ER stress upon acute viral infection. As a UPR target gene, the promoter region of the *RTN3* gene harbors binding sites for the ER-stress-related transcription factors ATF6 and CHOP. As a consequence, RTN3 upregulation inhibited RIG-I-mediated immune responses, but had only slight and minimal effects on MDA5- and TLR3-induced activation. We further demonstrated that RTN3 overexpression significantly attenuates the RIG-I-mediated production of proinflammatory cytokines.

In assessments of whether RIG-I is the predominant target through which RTN3 inhibits antiviral responses, our results demonstrated that RTN3 separately interacts with RIG-I and TRIM25 and that their interaction is augmented upon VSV stimulation. Furthermore, we showed that RTN3 overexpression impairs the K63-linked polyubiquitination of RIG-I in a TRIM25-dependent manner but does not interfere with RIG-I stability or interaction. Notably, the K63-linked polyubiquitination of TRIM25 was promoted while that of RIG-I was inhibited by RTN3 overexpression that led to RTN3 self-aggregation and the formation of a scaffold-like structure, indicating that RTN3 may interfere with the

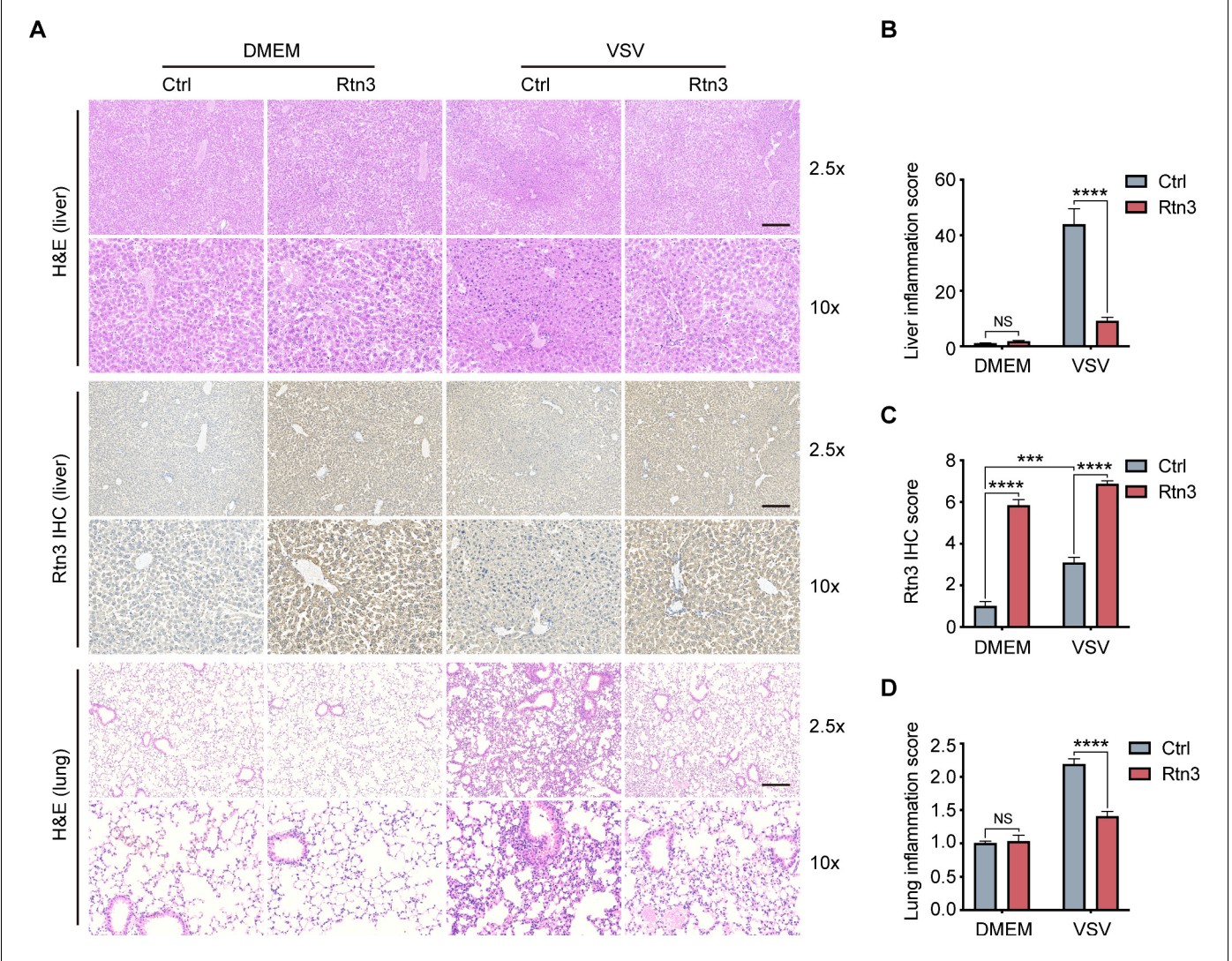

**Figure 7.** Rtn3 overexpression promotes inflammation resolution in mice. (**A**) Microcopy analysis of liver and lung tissue sections from the same mice as in (**Figure 5C**) with hematoxylin and eosin (H and E) staining, and HA-Rtn3 exotic expression was visualized by immunohistochemical staining (IHC). (**B**) Inflammation scores of liver tissue sections (n=4, 4, 4, 4) from the same mice shown in (**A**). The data were obtained and analyzed using ImageJ and visualized with GraphPad Prism 8. (**C**) IHC score assessment of liver tissue sections as in (A, middle), data was obtained and calculated with Image J plugin 'IHC Profiler' (**D**) Inflammation scores of lung tissue sections (n=4, 4, 4, 4) from the same mice shown in (**A**). The data were obtained and analyzed using ImageJ and visualized with GraphPad Prism 8. In (**B–D**), data are shown as the mean values ± SD (n = 4). *, p < 0.0332; **, p < 0.0021; ***, p < 0.0002; ****, p < 0.0001; by Sidak's multiple comparisons test.

The online version of this article includes the following source data and figure supplement(s) for figure 7:

**Source data 1.** Excel file of inflammation scores and description of statistical tests for panel B.

**Source data 2.** Excel file of IHC scores and description of statistical tests for panel C.

**Source data 3.** Excel file of inflammation scores and description of statistical tests for panel D.

**Figure supplement 1.** Viral infection in the liver increases upon RTN3 overexpression.

oligomerization of TRIM25 and its E3 ubiquitin ligase activity to promote RIG-I polyubiquitination. Furthermore, mapping analysis of truncated RTN3 constructs showed that RTN3 transmembrane domain 1 (TM1) is crucial for impairing RIG-I K63-linked polyubiquitination and inhibiting immune responses, indicating that a proper topological structure of RTN3 on the ER membrane is essential for its activity. Thus, the underlying mechanism of how RTN3 interferes with this process is worth further investigation.

Consistent with the in vitro results, in vivo experiments in mice showed that RTN3 is upregulated upon VSV infection or poly(I:C) stimulation. Importantly, our results provide evidence that RTN3 overexpression decreases the production of inflammatory factors and the neutrophil population in the liver and blood during RNA viral infection. The native neutrophil population may be primarily affected by the downregulation of cytokines and chemokines, and the circulating neutrophil population is affected secondarily. Because neutrophils have the capacity to damage tissue (*Kolaczkowska and Kubes, 2013*; *Ley et al., 2007*), neutrophil inhibition is crucial in the inflammatory response (*Ortega-Gómez et al., 2013*; *Soehnlein and Lindbom, 2010*). The suppression of chemokine CXCL8 (IL-8; *Figure 2—figure supplement 1H*) levels by RTN3 indicates that it may be a potential mediator of inflammation resolution by indirectly restricting neutrophil recruitment. Although the decreased macrophage populations in Rtn3-ovrerexpressing mice were not significant, we did observe a decreasing trend for these populations (*Figure 6—figure supplement 1C*). The histological results demonstrated that RTN3 overexpression attenuated the infiltration of lymphocytes and greatly relieved tissue inflammation and edema. Taken together, these results provide evidence that RTN3 functions to maintain tissue homeostasis and initiate the resolution of inflammation by inhibiting RIG-I-mediated antiviral responses and even terminating acute inflammation.

Therefore, we propose a working model that describes how RTN3 suppresses RIG-I-mediated antiviral innate immune responses (*Figure 8*). Upon RNA viral infection, the RIG-I signalosome undergoes TRIM25-mediated activation and induces the production of IFNs, cytokines and chemokines, leading to acute immune responses and inflammation. Subsequently, RTN3 is upregulated in an ER stress- and/or inflammation-dependent manner, which is directly associated with acute viral infection. Aggregated RTN3 on the endoplasmic reticulum interacts with TRIM25 and RIG-I, inhibiting the K63-linked polyubiquitination of RIG-I via the E3 ligase activity TRIM25 to block the RIG-I signaling cascade and attenuate innate immune responses. As a result, the production of IFNs,

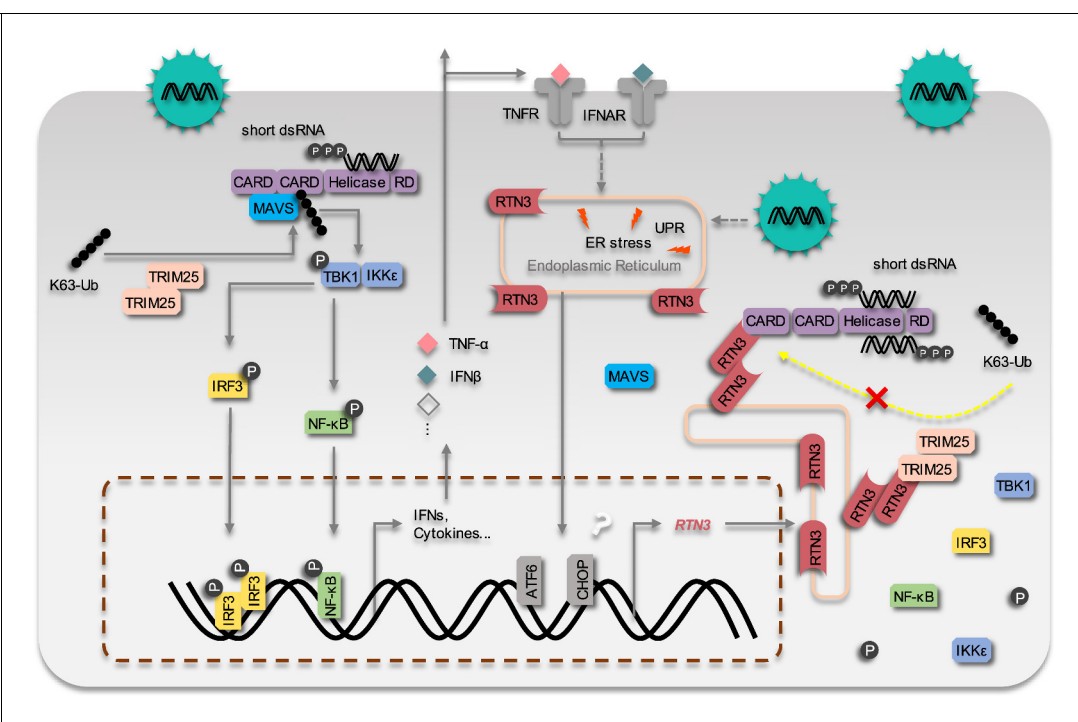

**Figure 8.** The working model of the inhibition of RIG-I mediated antiviral responses by RTN3 Upon RNA virus infection, the RIG-I signalosome is activated by TRIM25-mediated K63-linked polyubiquitination, and then triggers IRF3 and NF-κB activation and the expression of interferon, cytokines and inflammatory factors. On the other hand, RTN3 is up-regulated by acute viral infection and interferon/inflammatory factors in an ER stress-dependent manner. RTN3 is subsequently aggregated on the endoplasmic reticulum, interacts with RIG-I and TRIM25, rapidly diminishes the K63-linked polyubiquitination of RIG-I and hence suppresses antiviral immune and inflammatory responses. The inflammatory response is consequently mitigated and the resolution of acute inflammation is initiated.

cytokines and chemokines is attenuated, and the resolution of host inflammation is simultaneously initiated.

In summary, our present studies demonstrate that a conserved reticulon protein, RTN3, which is ubiquitously expressed in various tissues and organs, is upregulated upon RNA viral infection. In turn, upregulated RTN3 inhibits RIG-I signalosome activation by impairing the TRIM25-RIG-I axis. Our findings demonstrate a novel negative feedback mechanism for acute immune responses and inflammation resolution and the potential reconstitution of tissue homeostasis during viral infection.

## Materials and methods

### Mice and mouse experiments

All animal experiments were approved by the Institutional Animal Care and Use Committee of Sun Yat-Sen University, China. Wild-type (WT) C57BL/6 mice were purchased from VITAL RIVER, Beijing, China and used for the animal experiments. For Rtn3-induction models, 8-week-old female mice received an intravenous (I.V.) injected of poly(I:C) (400 μg/mouse) or an equal volume of phosphate-buffered saline (PBS) as controls or VSV-eGFP [$1 \times 10^7$ plaque-forming units (PFU)/g/mouse] for 12 hr. For RTN3-overexpressing mouse models, 8-week-old female mice were hydrodynamically injected with a plasmid encoding Rtn3 (HA-Rtn3) or the empty vector (HA-Ev). Twenty micrograms of DNA diluted in 2 ml of $1\times$ PBS was injected through the tail vein using a syringe with a 30-gauge needle. The injection was completed within 8–12 s, and 48 hr post DNA injection, the mice received an intraperitoneal (I.P.) injection of VSV-eGFP [$1 \times 10^7$ PFU/g/mouse] or an equal volume of DMEM for 12 hr.

### Cell culture

HEK293T, *RIG1* KO HEK293T (kindly provided by Dr. Yang Du [Zhongshan School of Medicine, Sun Yat-Sen University]), and HeLa cells were cultured in DMEM (Corning, 10–013-CV) supplemented with 10% fetal bovine serum (FBS) (Gibco, 10270–106), while A549 and THP-1 cells were cultured in RPMI 1640 medium (Corning, 10–040-CV) supplemented with 10% FBS. THP-1 cells stably overexpressing HA-RTN3 (THP-1$^{HA-RTN3}$) and control THP-1 cells (THP-1$^{HA-Ev}$) were cultured in RPMI 1640 medium supplemented with 10% FBS, whereas TRIM25 knockdown HEK293T cells (shTRIM25-HEK293T) were cultured in DMEM supplemented with 10% FBS. All cells were cultured in an incubator at 37°C under an atmosphere with 5% $CO_2$. HEK293T, A549, THP-1, and HeLa cells were obtained from ATCC (American Type Culture Collection). Identities of the cell lines used in this study were authenticated by STR profiling. Mycoplasma contamination status was tested as negative with Myco-Lumi Luminescent Mycoplasma Detection Kit (Beyotime, C0298M).

Peripheral blood mononuclear cells (PBMCs) were isolated from whole blood of healthy donors that was purchased from Guangzhou Blood Bank, using Lymphoprep (StemCell Technologies, Vancouver, BC) according to the manufacturer's instructions. The PBMCs were cultured in RPMI 1640 containing 10% FBS at a final concentration of $1\times10^6$ cells/mL. All human blood works were approved by the medical ethics committee at Sun Yat-sen University.

### Antibodies and reagents

The following antibodies were used in this study: anti-RTN3 (BA3533), purchased from BOSTER; anti-HA (M132-3), purchased from MBL; anti-Flag (AE005), anti-GFP (AE0122), were purchased from ABclonal; anti-β-actin (HC201-01), purchased from TransGen; anti-TRIM25 (67314–1-Ig), purchased from Proteintech; anti-GST (#2622), anti-Phospho- IKKα/β (#2697), anti-Phospho-TBK1 (#5483P), anti-TBK1 (#3504), anti-Phospho-IRF3 (#4947), anti-IRF3 (#4302), anti-Phospho-NF-κB p65 (#3033), anti-NF-κB p65 (#8242), anti-K63-linkage Specific Polyubiquitin (#5621), and anti-Ubiquitin (#43124) were purchased from Cell Signaling Technology; goat anti-mouse IgG IRDye680RD (C90710-09) and goat anti-rabbit IgG IRDye800CW (C80925-05) were purchased from Li-COR. Following reagents were used in this study: Polyinosinic–polycytidylic acid potassium salt [poly (I:C)] (P9852, average MW: 200,000–500,000), purchased from Sigma-Aldrich; Human recombinant TNF-alpha protein (10602-H01H), purchased from SinoBiological; Lipofectamine 2000, purchased from Invitrogen; anti-GFP beads (KTSM1301), purchased from Alpa-life; Glutathione Sepharose 4B (17-0756-01),

purchased from GE Healthcare; The BD CBA Mouse Inflammation Kit (552364), purchased from BD Bioscience.

## Viral titer and infection

VSV-eGFP and VSV (kindly provided by Dr. Meng Lin, School of Life Science) were propagated and amplified in HEK293T cells and titrated on Vero cells with a standard plaque assay. SeV was a gift from Dr. Yang Du. For the in vivo study, VSV-eGFP was injected into mice at a titer of $1 \times 10^7$ plaque-forming units (PFU)/g for 12 hr. For the in vitro study, cells were infected with VSV-eGFP (MOI = one for RTN3 induction and immune response activation; MOI = 0.05 for fluorescence and phase contrast (PH) analyses) or SeV (MOI = 0.1 for immune response activation).

## Plasmids and transfection

The plasmids used in the present study were constructed as follows. *RTN3* and *TRIM25* were obtained from the HEK293T cDNA library and cloned into the eukaryotic expression vector pcDNA3.1 in-frame with an HA or Flag tag as well as the eukaryotic expression vectors pEGFP-C2 and pEBG. The HA-tagged *RTN3* fragment was also subcloned into the pMSCV-PURO retrovirus expression vector, and the *RTN3* fragment was truncated based on its structure and subcloned into pcDNA3.1 to generate the HA-tagged mutants T1, T2, T3, and T4. The Flag-RIG-I, Flag-RIG-I CARDs [RIG-I (N)], Flag-TLR3, Flag-TBK1, Flag-P65, Flag-MDA5, Flag-IRF3, HA-tagged Ub, and K63-Ub (K63 only) were kindly provided by Dr. Yang Du. The RIG-I fragment was also subcloned into pEGFP-C2. PrimeSTAR® MAX DNA Polymerase (Takara, RO45A) and a ClonExpressII One Step Cloning Kit (Vazyme, C11-02) were used to generate all the constructs. For transfection, HEK293T cells were seeded into 24- or 6-well plates overnight and then transfected using Lipofectamine 2000 according to the manufacturer's protocol. Poly(I:C) was transfected into HEK293T cells using Lipofectamine 2000.

## Generation of gene-modified cell lines

HEK293T cells were seeded into six-well plates at a density of $0.5 \times 10^6$ cells/ml and incubated overnight. Then, 2.5 ml/well of a *TRIM25* shRNA-encoding lentivirus was added to the plate, which was then placed in an incubator. Twenty-four hours postinoculation, the cells were infected with a second batch of virus for another 24 hr. The infected cells were then reseeded into six-well plates with fresh RPMI 1640 medium supplemented with 10% FBS and cultivated for 36 hr. Puromycin (2–4 μg/ml, Thermo Fisher, A113803) was used for TRIM25-knockdown HEK293T cell selection, and immunoblot analysis was performed to determine the knockdown efficiency. For TRIM25 knockdown-HEK293T cells generation, the following TRIM25 shRNA sequences were used: (1), 5'- CCGGAACAGTTAG TGGATTTA-3'; (2), 5'-GAACTGAACCACAAGCTGATA-3'. Both sequences target the coding region exon1 of TRIM25 gene and cloned into pLKO.1 lentivirus expression vector. The same methods were performed for transient *RTN3* knockdown with the following shRNA sequences: (1), 5'- CCCGA TTGTCTATGAGAAGTA-3'; (2), 5'-CTTTGGCACCACGCTGATCAT-3'.

## Luciferase reporter assays

HEK293T cells were seeded in 24-well plates overnight and then transfected using Lipofectamine 2000 with 100 ng of an ISRE-luciferase reporter (firefly luciferase), 100 ng of an IFNB luciferase reporter (firefly luciferase), or 100 ng of an NF-κB luciferase reporter (firefly luciferase) together with 20 ng pRL-TK plasmid (Renilla luciferase), 150 ng Flag-RIG-I (N), Flag-MDA5, or Flag-IRF3-5D mutant and increasing amounts of HA-RTN3. Twenty-four hours post transfection, the cells were collected, and luciferase activity was measured using a Dual-Luciferase Assay kit (Promega, E1960) with a Synergy2 Reader (Bio-Tek) according to the manufacturer's protocol. The relative level of gene expression was determined by normalizing firefly luciferase activity to Renilla luciferase activity. Poly(I:C) (5 μg/ml) or SeV (MOI = 0.1) was also used as for activation in the assay without RIG-I (N) construct transfection.

## Real-time PCR

Total RNA was extracted from HEK293T or A549 cells dissolved in TRIzol reagent (Invitrogen, 15596026) and then reverse transcribed into cDNA using HiScript III RT SuperMix (Vazyme). A SYBR

Green I Master Mix kit (Roche, 04887352001) was used for real-time PCR with a LightCycler 480 system (Roche). The following primers were used for the assay: Human *RTN3* forward, 5'-AGCTGGCTG TCTTCATGTGG-3'; reverse, 5'-CTCGGGCGATGCCAACATAG-3'. Human *ACTB* forward, 5'-CACCA TTGGCAATGAGCGGTTC-3'; reverse, 5'-AGGTCTTTGCGGATGTCCACGT-3'. Mouse *Rtn3* forward, 5'-AGCTGGCTGTCTTCATGTGG-3'; reverse, 5'-TCCCGGGCAATCCCAACA-3'. Mouse *Gapdh* forward, 5'-AGGCCGGTGCTGAGTATGTC-3'; reverse, 5'-TGCCTGCTTCACCACCTTCT-3'. Human *IFNB1*, forward, 5'-CCTACAAAGAAGCAGCAA-3'; reverse 5'-TCCTCAGGGATGTCAAAG-3'. Human *IFIT1* forward, 5'-GCCTTGCTGAAGTGTGGAGGAA-3'; reverse, 5'-ATCCAGGCGATAGGCAGAGA TC-3'. Human *IFIT2* forward, 5'-GGAGCAGATTCTGAGGCTTTGC-3'; reverse, 5'-GGATGAGGC TTCCAGACTCCAA-3'. Human *TNF* forward, 5'-CTCTTCTGCCTGCTGCACTTTG-3'; reverse, 5'-A TGGGCTACAGGCTTGTCACTC-3'. Human *IL6* forward, 5'-AGACAGCCACTCACCTCTTCAG-3'; reverse, 5'-TTCTGCCAGTGCCTCTTTGCTG-3'. Human *CCL20* forward, 5'-AAGTTGTCTGTG TGCGCAAATCC-3'; reverse, 5'-CCATTCCAGAAAAGCCACAGTTTT-3'. Other primers used in the array were listed in *Figure 2—figure supplement 1—source data 9*.

## Coimmunoprecipitation and immunoblot analysis

For coimmunoprecipitation assays, cells transfected and stimulated with the appropriate ligands were lysed with IP buffer (Beyotime, P0013), after which whole cell extracts were collected and incubated with anti-GFP beads or Glutathione Sepharose 4B at 4°C for 1 hr or overnight. Then, the beads were washed 4–5 times with IP buffer, and the immunoprecipitants were eluted with 2× SDS loading buffer and then resolved by SDS-PAGE. Subsequently, the proteins were transferred to PVDF membranes (Millipore, ISEQ00010) and further incubated with the indicated primary and secondary antibodies. The images were visualized using an Odyssey Sa system (LI-COR).

## Confocal microcopy

WT HeLa cells were seeded in 8-well Millicell@EZ slides (Millipore Ltd, PEZGS0816) overnight and then transfected using Lipofectamine 2000. Twenty-four hours posttransfection, the cells were carefully washed with 1× PBS three times (2 ml/well, 5 min/time) and then fixed with 4% PFA for 15 min at 4°C. After the cells were penetrated with 0.2% Triton X-100 at room temperature for 5 min, they were washed and blocked with 3% BSA in 1× PBS for 1 hr at 4°C and then assayed using the indicated primary and secondary antibodies. Microcopy was performed with a ZESS LSM800 system, and images were captured and processed using ZESS ZEN Imaging Software.

## Histology

Infected mice were sacrificed at the indicated timepoints, and their livers and lungs were dissected and fixed in 4% PFA. After embedding, paraffin sections were stained with hematoxylin and eosin (H and E) or subjected to standard immunohistochemical staining (IHC) before being visualized by bright field microscopy.

## Statistical analyses

The data are presented as the means ± SD (for in vivo experiments, the values are presented as the means ± SD of n mice), and Sidak's multiple comparisons test was used for all statistical analyses performed with GraphPad Prism 8. Differences between two groups were considered significant at p value < 0.0332.

## Acknowledgements

We thank all the members of our laboratory for their critical assistance and helpful discussions. This work is supported by grants from the Natural Science Foundation of China (81871643 and 32061143008) to EK and the Natural Science Foundation of China (81971928) to XL.

## Additional information

### Funding

| Funder | Grant reference number | Author |
|---|---|---|
| National Natural Science Foundation of China | 81871643 | Ersheng Kuang |
| National Natural Science Foundation of China | 32061143008 | Ersheng Kuang |
| National Natural Science Foundation of China | 81971928 | Xiaojuan Li |

The funders had no role in study design, data collection and interpretation, or the decision to submit the work for publication.

### Author contributions

Ziwei Yang, Formal analysis, Investigation, Methodology, Writing - original draft; Jun Wang, Bailin He, Formal analysis, Methodology; Xiaolin Zhang, Investigation; Xiaojuan Li, Conceptualization, Supervision, Funding acquisition; Ersheng Kuang, Conceptualization, Supervision, Funding acquisition, Project administration, Writing - review and editing

### Author ORCIDs

Ziwei Yang (iD) https://orcid.org/0000-0003-3362-0946
Ersheng Kuang (iD) https://orcid.org/0000-0002-4976-3311

### Ethics

Human subjects: All human samples (PBMCs) were anonymously coded in accordance with the local ethical guidelines. The Ethics Review Board of Zhongshan School of Medicine, Sun Yat-Sen University approved this study. Approval No: ZhongshanSchool of Medicine Ethics [2021] NO.016.
Animal experimentation: All animal experiments were approved by the Institutional Animal Care and Use Committee of Sun Yat-Sen University, China. Approval No: SYSU-IACUC-2021-B0025.

### Decision letter and Author response

Decision letter https://doi.org/10.7554/eLife.68958.sa1
Author response https://doi.org/10.7554/eLife.68958.sa2

## Additional files

### Supplementary files

• Transparent reporting form

### Data availability

All data generated or analysed during this study are included in the manuscript and supporting files.

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
