## [Decision Letter]

**Acceptance summary:**

This study is of broad interest to readers in the field of antiviral innate immunity and those studying inflammation and the molecular mechanisms that resolve inflammation. Kuang and colleagues identify and characterize a new regulatory protein that helps to maintain immune homeostasis following activation of the cytoplasmic RNA sensor RIG-I. The authors present a combination of in vivo and in vitro data that highlight a delicate regulatory circuit in innate immunity to RNA virus infection.

**Decision letter after peer review:**

Thank you for submitting your article "RTN3 inhibits RIGI-I-mediated antiviral responses by impairing TRIM25-mediated K63-linked polyubiquitination" for consideration by *eLife*. Your article has been reviewed by 2 peer reviewers, and the evaluation has been overseen by a Reviewing Editor and Sara Sawyer as the Senior Editor. The reviewers have opted to remain anonymous.

Essential Revisions:

1) Functional analyses in RTN3 knockdown or knockout cells, ideally in primary cells.

2) Mechanistic and interaction studies with endogenous RTN3 to confirm the proposed molecular mechanism.

3) Addressing the specific issues about controls that have been pointed out by reviewers.

*Reviewer #1:*

In this manuscript "RTN3 inhibits RIG-I-mediated antiviral responses by impairing TRIM25-mediated K63-linked polyubiquitination", Yang et al. report that the reticulon member RTN3 is upregulated upon VSV infection or poly(I:C) stimulation, and that its interaction with the E3 ligase TRIM25 and the sensor RIG-I impairs innate immune signaling to maintain immune homeostasis and resolve inflammation. The authors show that this interaction inhibits the K63-linked ubiquitination of RIG-I mediated by TRIM25, which is a process necessary for RIG-I signaling. The upregulation of RTN3 expression following RNA virus infection was confirmed in vivo, and the authors further demonstrated that neutrophil numbers, lymphocyte infiltration, and inflammation including edema were reduced in mice in which RTN3 was overexpressed. Altogether, this study identifies RTN3 as a regulatory protein that controls RNA virus infection associated upregulation of the innate immune/inflammatory response in order to alleviate tissue inflammation.

The conclusions of this paper are overall supported by the data, and the experiments which represent both in vitro mechanistic studies and in vivo functional analysis, are well designed. The study demonstrates that enforced RTN3 expression inhibits RIG-I mediated innate immune responses and alleviates inflammation. However, additional extensive analyses detailing the effect of RTN3 gene targeting on antiviral innate immunity, as well as additional experiments with endogenously expressed RTN3, are needed to strengthen the physiological role of RTN3 in inflammation/immunity.

Major Strengths:

The authors performed a very thorough characterization of RTN3 and its mechanism. They convincingly show binding of RTN3 to both TRIM25 and RIG-I, and the inhibitory effect that this interaction mode has on K63-linked ubiquitination and antiviral signaling of RIG-I. It is clear from the in vitro and in vivo data that endogenous RTN3 is upregulated upon VSV infection or poly(I:C) stimulation, and that its enforced expression dampens the activation of downstream signaling mediators and the induction of proinflammatory genes. The authors performed detailed mapping studies to identify the domain responsible for RTN3's ability to block antiviral signaling, which suggested that proper localization of RTN3 on ER membranes is required for RTN3's immune dampening activity. The in vivo data convincingly show upregulation of endogenous Rtn3 upon VSV infection, and that the neutrophil population is decreased and inflammation is alleviated when Rtn3 expression is enforced.

The authors utilized a number of cell lines to support their findings, and numerous assays to validate their hypothesis and proposed model.

Weaknesses:

The design of the experiments solely relies on RTN3 overexpression. The study needs to be further strengthened by experiments that analyze: (a.) endogenous RTN3/TRIM25/RIG-I interaction or complex formation during virus infection or dsRNA stimulation; and (b.) the effect of RTN3 knockdown or knockout on RIG-I antiviral/proinflammatory signaling. RTN3 gene-targeting analysis in primary cells (human or mouse cells) would be preferred, if possible.

The overall writing and structure of the manuscript could be further improved for the reader's benefit.

1. Most of the experiments are done with RTN3 overexpression, which clearly showed an immune dampening effect. What is the effect of RTN3 knockdown or knockout on RIG-I signaling and proinflammatory gene induction in response to virus infection or poly(I:C) stimulation? Such gene-targeting data, especially if performed in primary cells (human or mouse cells) if possible, would greatly strengthen the physiological role of RTN3.

2. The authors performed extensive interaction and mechanistic studies with overexpressed RTN3, which convincingly showed that RTN3 binds to both TRIM25 and RIG-I and blocks the TRIM25-mediated K63-linked ubiquitination of RIG-I. Data assessing endogenous TRIM25-RTN3 binding/co-localization during infection or poly(I:C) stimulation, and the effect of RTN3 silencing on RIG-I signaling would further bolster the proposed mechanism.

3. Figure 3H: There is not an appreciable effect of HA-RTN3 on K63-ubiquitination of TRIM25. Overall, it is unclear how enhanced TRIM25 K63-ubiquitination/auto-ubiquitination by RTN3 would lead to suppressed TRIM25 catalytic activity and thereby dampened TRIM25-mediated RIG-I K63-ubiquitination. Overall, this part of the mechanism is not very clear. This reviewer suggests moving Figure 3H to the Supplementary Data, or the authors may further discuss this.

4. Figure 4B (IP, right 3 lanes): Although there was no effect observed for HA-RTN3 on RIG-I ubiquitination in the absence of TRIM25, it is unclear why the overall RIG-I ubiquitination signal in shTRIM25 cells is as high as that in shCtrl cells. One would expect overall reduced RIG-I K63-ubiquitination when TRIM25 is silenced, according to reduced downstream signaling in shTRIM25 cells as shown in Figure 4A. This reviewer is wondering whether the reason for the still-high RIG-I ubiquitination in shTRIM25 cells could be because the authors blotted the IP with a pan-Ub antibody, which would also recognize K48-linked ubiquitination of RIG-I, but not with HA antibody that would specifically recognize the overexpressed HA-tagged K63-Ub. The authors should repeat this experiment and blot the IP with HA antibody instead of Ub antibody (here the authors could use a differently-tagged RTN3)

*Reviewer #2:*

In this manuscript Yang et al., explore the potential role of RTN3, a member of the reticulon family, in inhibition of RIG-I and TRIM25 -mediated responses. The authors show that RTN3 expression is induced during Vesicular Stomatitis virus (VSV) infection or double stranded RNA (dsRNA) stimulation, as well as TNF stimulation. Overexpression of RTN3 reduces IFNbeta and NF-κB responses upon stimulation with RIG-I (N) or VSV infection. RTN3 interacts with RIG-I and TRIM25 and is proposed that RTN3 inhibits K63-linked polyubiquitination of RIG-I resulting in inhibition of inflammation. The authors performed mouse in vivo experiments with overexpression of RTN3, which show reduced neutrophil infiltration. The authors conclude that RTN3 is a "proresolving mediator' of immune and inflammatory responses.

The main strength of this manuscript is the effort in performing in vivo mouse experiments to elucidate in a closer physiological setting the potential role of RTN3 in immunity and inflammatory responses, although the overexpression approach reduces the significance of the findings.

The weaknesses of the manuscript include the very heavy reliance on overexpression assays and the complete lack of RTN3 knockdown or knockout experiments to confirm phenotypes. The overexpression experiments also have the weakness that may not represent the real levels of RTN3 and these levels may never be reached during infection. Additional weaknesses include lack of important controls, overstatements that are not supported by the data, lack of analysis of endogenous ubiquitination, and a model/conclusion that seems not very well supported by the data.

Overall, these studies could be of importance and significant to the field if experiments are designed to address more closely the endogenous role of RTN3. Due to the lack of endogenous analysis and reliance of overexpression assays, the study is likely to have only moderate impact in the field.

Specific issues:

1) Figure 1D – why do the authors focus on TNF as the potential cytokine that induces RTN3 expression? What about type-I IFNs?

2) Figure 2A, Luciferase reporter assays: these assays are not well designed to address the role of RTN3 upon overexpression of RIG-I (N). These experiments are biased towards inhibitory effects of the overexpressed protein. RIG-I (N) needs to be carefully titrated in the absence of RTN3.

3) Figure 3D, and S1F, immunofluorescence (colocalization) experiments: need controls with each protein overexpressed alone. Also, for S1F, the authors claim there is aggregation in the endoplasmic reticulum. There is no evidence or markers of localization in the ER. This must be based on the known localization of RTN3, however, the RIG-I and TRIM25 have been proposed to localized to other compartments, including the mitochondria and in some cases in the Golgi or in mitochondrial-associated endoplasmic reticulum membranes (MAMs). Without more evidence of localization it is not clear where RTN3 localizes with RIG-I/TRIM25.

4) Lines 184 -188: "Since TRIM25 directly targets RIG-I, the coimmunoprecipitation assays were performed in RIG-I knock out HEK293T cells and the result showed that TRIM25-RTN3 interaction did not required RIG-I, we thus concluded that RTN3 directly binds to TRIM25 and excluded the possibility that RTN3 may interacts with TRIM25 through the interaction between TRIM25 and RIG-I (Figure S3 C)". : This statement is not correct. The fact that TRIM25 and RTN3 interact in RIG-I knockout cells doesn't mean that the interaction between RTN3 and TRIM25 is "direct". The interactions is independent of RIG-I, but the authors need to do in vitro binding assays using purified proteins if they want to conclude that interactions are "direct". TRIM25 has other interacting factors besides RIG-I.

5) Figures 3G, 3H and S3E, ubiquitination assays: there are a few issues with these experiments.

A) there is extensive background of Ubiquitin in the asbcence of the immunoprecipitated protein (lane 1 in figures 3G and 3H, in the IP and the IB for Ub).

B) these experiments are performed with overexpressed HA-Ub-K63, but the immunoblot is performed with anti-Ub. The immunoblot is not specific for HA-Ub-K63. In addition, these experiments do not show the covalently modified form of ubiquitinated RIG-I (or TRIM25 in figure 3H).

C) All these experiments need to show the levels of overexpressed HA-Ub-K63 in the WCL to control for the transfection efficiencies.

D) these experiments should also be performed without overexpression of ubiquitin to determine the endogenous ubiquitination of RIG-I

6) The conclusion that RTN3 is a "proresolving mediator' of immune and inflammatory responses, cannot be made when using only overexpression assays. The levels of RTN3 that are present in overexpression assays are not physiological and most likely do not represent real conditions. 'proresolving' mediator of inflammation is also an odd definition of these observation s. RIG-I has multiple negative regulators, which keep the pathway from overactivation. There is really no evidence that RTN3 acts at the 'resolving' stage of inflammation.

7) The model shown in figure S7. This implies that there are two different cells, one that is undergoing 'inflammation' and another different cell that is 'resolving' inflammation. This also is not really supported by any data. The authors would need to do single-cell an analysis of infected vs non-infected cells, as well as more detailed analysis of which cytokines are controlling inflammation or dampening neutrophil numbers. The 'resolving' cell could have nothing to do with infection, and could be responding to cytokines. In addition, this conclusion would also need the use of knockout mice.

---

## [Author Response]

1. Most of the experiments are done with RTN3 overexpression, which clearly showed an immune dampening effect. What is the effect of RTN3 knockdown or knockout on RIG-I signaling and proinflammatory gene induction in response to virus infection or poly(I:C) stimulation? Such gene-targeting data, especially if performed in primary cells (human or mouse cells) if possible, would greatly strengthen the physiological role of RTN3.

Thank you for your suggestion. Here, we performed RTN3 knockdown experiments in PBMCs (human normal primary cells) and provided a new figure (Figure 3) that shows that RTN3 knockdown positively manipulated antiviral immune and inflammatory responses. As shown in this figure, the phosphorylation levels of TBK1, p65 and IRF3 were augmented substantially during VSV infection (Figure 3B), leading to the enhanced transcription of interferon and proinflammatory factors (Figure 3C), along with the significantly compromised antiviral responses toward viral infection (Figure 3D). Furthermore, we also provided results showing that RTN3 knockdown significantly upregulates K63-linked polyubiquitination of RIG-I under VSV infection (Figure 4—figure supplement 1F,1G).

“RTN3 depletion augments antiviral immune responses

To further elucidate the physiological function of RTN3 in antiviral responses, we depleted its expression by short hairpin RNAs (shRNAs) in human healthy donor PBMCs (Figure 3A), in which RTN3 normally undergoes upregulation during viral infection (Figure 1F). Within our experience, complete or vast obliteration of the expression of RTN3 by knockout or knockdown leads to cell death; hence, RTN3 expression were moderately depleted with transient shRNA transduction (Figure 3A). PBMCs were infected with control or RTN3 shRNA-encoding lentivirus for 12 h, and the infected cells were then settled for the following 24 h and subsequently infected with VSV-eGFP. Consistent with the decreased phosphorylation levels of TBK1, p65 and IRF3 in RTN3 overexpressing cells, RTN3 depletion significantly augmented the phosphorylation and activation of these critical kinases and transcription factors upon RNA viral infection (Figure 3B). Further RT-PCR analyses determined that the levels of IFNB1, IFIT2, IFIT1, TNF and the proinflammatory cytokines CCL20 and IL6 were enhanced by RTN3 knockdown during VSV infection (Figure 3C), while the amplification of VSV was compromised (Figure 3D). Collectively, depressed RTN3 expression facilitates antiviral innate immune and inflammatory responses upon viral infection.”

And (Page 7, Line 236-237):

“In contrast, RTN3 depletion enhanced the K63-linked polyubiquitination level of RIG-I (Figure 4—figure supplement 1F, 1G).”

2. The authors performed extensive interaction and mechanistic studies with overexpressed RTN3, which convincingly showed that RTN3 binds to both TRIM25 and RIG-I and blocks the TRIM25-mediated K63-linked ubiquitination of RIG-I. Data assessing endogenous TRIM25-RTN3 binding/co-localization during infection or poly(I:C) stimulation, and the effect of RTN3 silencing on RIG-I signaling would further bolster the proposed mechanism.

As the reviewer suggested, we performed IP analyses in human PBMCs using anti-RTN3 antibody and IgG isotype. Consistent endogenous interactions (RTN3 interacts with both TRIM25 and RIG-I) were observed in PBMCs, and the interaction was moderately strengthened upon VSV stimulation (Figure 4F). Furthermore, the enhanced K63 ubiquitination of RIG-I was also observed under RTN3 depletion (Figure 4—figure supplement 1F, 1G). In addition, with regard to the perturbation of RIG-I signaling when RTN3 was knocked down, we also proved that depletion of RTN3 enhanced the antiviral immune and inflammatory responses after virus infection (Figure 3B, 3C).

The following description has been added to the manuscript (Page 6-7, Lines 218-220):

“Immunoprecipitation using anti-RTN3 antibody in human PBMCs infected with VSV demonstrated endogenous interaction between RTN3 and TRIM25/RIG-I (Figure 4F)”

And (Page 7, Line 236-237):

“In contrast, RTN3 depletion enhanced the K63-linked polyubiquitination level of RIG-I (Figure 4—figure supplement 1F, 1G).”

3. Figure 3H: There is not an appreciable effect of HA-RTN3 on K63-ubiquitination of TRIM25. Overall, it is unclear how enhanced TRIM25 K63-ubiquitination/auto-ubiquitination by RTN3 would lead to suppressed TRIM25 catalytic activity and thereby dampened TRIM25-mediated RIG-I K63-ubiquitination. Overall, this part of the mechanism is not very clear. This reviewer suggests moving Figure 3H to the Supplementary Data, or the authors may further discuss this.

Thank you for your comments and suggestion. We repeated these experiments, including both K63-linked polyubiquitination and WT polyubiquitination of TRIM25 mediated by overexpression of RTN3, and we detected the ubiquitination level with an anti-HA antibody but not an anti-ubiquitin antibody. A clearer tendency in the promotion of TRIM25 K63-linked polyubiquitination can now be observed (Figure 4—figure supplement 1H, 1I), probably because RTN3 might damper the delivery of K63-ubiquitin form TRIM25 to RIG-I. The fine-tuned manner of how RTN3 manipulates TRIM25 on its catalytic activity in RIG-I ubiquitination and activation is critical and of great interest; however, it is another story, and we will further explore the mechanism in our new projects.

As the reviewer and editors suggested, we have moved these results to the supplementary data (Figure 4—figure supplement 1).

4. Figure 4B (IP, right 3 lanes): Although there was no effect observed for HA-RTN3 on RIG-I ubiquitination in the absence of TRIM25, it is unclear why the overall RIG-I ubiquitination signal in shTRIM25 cells is as high as that in shCtrl cells. One would expect overall reduced RIG-I K63-ubiquitination when TRIM25 is silenced, according to reduced downstream signaling in shTRIM25 cells as shown in Figure 4A. This reviewer is wondering whether the reason for the still-high RIG-I ubiquitination in shTRIM25 cells could be because the authors blotted the IP with a pan-Ub antibody, which would also recognize K48-linked ubiquitination of RIG-I, but not with HA antibody that would specifically recognize the overexpressed HA-tagged K63-Ub. The authors should repeat this experiment and blot the IP with HA antibody instead of Ub antibody (here the authors could use a differently-tagged RTN3).

According to the suggestion of the reviewer, we repeated this experiment with Flag-tagged RTN3 and detected the ubiquitination level with an anti-HA antibody. The new result shows that the overall ubiquitination level is inferior in shTRIM25 cells compared with shCtrl cells (Figure 5B), which is consistent with the weaker phosphorylation levels of TBK1, P65 and IRF3, as shown in Figure 5A.

Previous studies have reported that TRIM4 and MEX3C are two positive regulators of RIG-I-mediated antiviral immune responses, both of which could significantly promote the K63-linked polyubiquitination of RIG-I upon virus infection (Kuniyoshi et al., 2014; Yan et al., 2014). We speculate that the remaining K63-linked polyubiquitination of RIG-I in TRIM25 knockdown cells was possibly caused by the compensatory effect of other regulators, such as TRIM4 or MEX3C, upon VSV infection.

Reviewer #2:In this manuscript Yang et al., explore the potential role of RTN3, a member of the reticulon family, in inhibition of RIG-I and TRIM25 -mediated responses. The authors show that RTN3 expression is induced during Vesicular Stomatitis virus (VSV) infection or double stranded RNA (dsRNA) stimulation, as well as TNF stimulation. Overexpression of RTN3 reduces IFNbeta and NF-κB responses upon stimulation with RIG-I (N) or VSV infection. RTN3 interacts with RIG-I and TRIM25 and is proposed that RTN3 inhibits K63-linked polyubiquitination of RIG-I resulting in inhibition of inflammation. The authors performed mouse in vivo experiments with overexpression of RTN3, which show reduced neutrophil infiltration. The authors conclude that RTN3 is a "proresolving mediator' of immune and inflammatory responses.The main strength of this manuscript is the effort in performing in vivo mouse experiments to elucidate in a closer physiological setting the potential role of RTN3 in immunity and inflammatory responses, although the overexpression approach reduces the significance of the findings.The weaknesses of the manuscript include the very heavy reliance on overexpression assays and the complete lack of RTN3 knockdown or knockout experiments to confirm phenotypes. The overexpression experiments also have the weakness that may not represent the real levels of RTN3 and these levels may never be reached during infection. Additional weaknesses include lack of important controls, overstatements that are not supported by the data, lack of analysis of endogenous ubiquitination, and a model/conclusion that seems not very well supported by the data.Overall, these studies could be of importance and significant to the field if experiments are designed to address more closely the endogenous role of RTN3. Due to the lack of endogenous analysis and reliance of overexpression assays, the study is likely to have only moderate impact in the field.Specific issues:1) Figure 1D – why do the authors focus on TNF as the potential cytokine that induces RTN3 expression? What about type-I IFNs?

As described in our manuscript, the detection of the transcriptional level of *TNF* was used to monitor the effectiveness of VSV-eGFP infection or poly(I:C) treatment toward RTN3 expression and inflammation. We observed a similar upregulation tendency of *TNF* and *RTN3* expression during these processes; thus, we detected the relationship between TNF-α and RTN3 expression and found that TNF-α can induce RTN3 expression. We also tested the effect of IFN-β on RTN3 induction and found that IFN-β exhibits a similar induction of RTN3 expression. We have added the data and shown that IFN-β induces RTN3 upregulation (Figure 1D bottom and Figure 1E bottom**)**.

The following description has been revised in the manuscript (Page 4, lines 120-122):

“Interestingly, we observed that both TNF-α and IFN-β could upregulate RTN3 expression at both the protein (Figure 1D) and mRNA levels (Figure 1E)”.

2) Figure 2A, Luciferase reporter assays: these assays are not well designed to address the role of RTN3 upon overexpression of RIG-I (N). These experiments are biased towards inhibitory effects of the overexpressed protein. RIG-I (N) needs to be carefully titrated in the absence of RTN3.

Thank you for your suggestion. We repeated these experiments with optimized amounts of RIG-I(N) and RTN3. In our new data, we showed dose-dependent inhibition of ISRE-luc, IFNβ-luc and NF-κB-luc by RTN3 (Figure 2A).

3) Figure 3D, and S1F, immunofluorescence (colocalization) experiments: need controls with each protein overexpressed alone. Also, for S1F, the authors claim there is aggregation in the endoplasmic reticulum. There is no evidence or markers of localization in the ER. This must be based on the known localization of RTN3, however, the RIG-I and TRIM25 have been proposed to localized to other compartments, including the mitochondria and in some cases in the Golgi or in mitochondrial-associated endoplasmic reticulum membranes (MAMs). Without more evidence of localization it is not clear where RTN3 localizes with RIG-I/TRIM25.

According to the reviewer’s suggestion, we have added images of the ctrl groups, including GFP-TRIM25, GFP-RIG-I and mCherry-RTN3, separately to the main panel (the previous Figure 3D, now renamed Figure 4D). In revised Figure 1—figure supplement 1G, we also added images of mCherry-Ev with/without poly(I:C) stimulation.

To strengthen our conclusion that RTN3 aggregates in the endoplasmic reticulum, we supplemented the ER marker calnexin in confocal IF images, which clearly showed the colocalization of RTN3 and calnexin within the aggregates after poly(I:C) stimulation (Figure 1—figure supplement 1I). These images clearly confirmed the known localization of RTN3 in the ER, either non-aggregated or aggregated, as well as RTN3 colocalization with RIG-I/TRIM25 in the ER.

The following description has been added to the manuscript (Page 4-5, lines 136-138):

“It was further confirmed that RTN3 was colocalized with calnexin on the ER as well as their tight colocalization within the aggregation under poly(I:C) stimulation, which substantiated that RTN3 aggregation was localized on the endoplasmic reticulum (ER) (Figure 1—figure supplement 1I).”

4) Lines 184 -188: "Since TRIM25 directly targets RIG-I, the coimmunoprecipitation assays were performed in RIG-I knock out HEK293T cells and the result showed that TRIM25-RTN3 interaction did not required RIG-I, we thus concluded that RTN3 directly binds to TRIM25 and excluded the possibility that RTN3 may interacts with TRIM25 through the interaction between TRIM25 and RIG-I (Figure S3 C)". This statement is not correct. The fact that TRIM25 and RTN3 interact in RIG-I knockout cells doesn't mean that the interaction between RTN3 and TRIM25 is "direct". The interactions is independent of RIG-I, but the authors need to do in vitro binding assays using purified proteins if they want to conclude that interactions are "direct". TRIM25 has other interacting factors besides RIG-I.

Thank you for your comment and suggestion. We agree that this is an imprecise statement, and the description of “direct interaction” requires data from an in vitro pulldown assay. Considering that the specific transmembrane domains of RTN3 inserted in the endoplasmic reticulum membrane are required for the interaction with TRIM25/RIG-I and its inhibitory function, as elucidated in Figure 5 C, 5D and Figure 5—figure supplement 1, we speculated that the interaction between RTN3 and TRIM25 or RIG-I would need a certain condition. Thus, we have replaced our statement “…directly binds…” with “…independent on RIG-I”.

The following description has been adjusted in the manuscript (Page 6, lines 211-212):

“thus we concluded that RTN3 binds to TRIM25 independent of RIG-I…”

5) Figures 3G, 3H and S3E, ubiquitination assays: there are a few issues with these experiments.A) There is extensive background of Ubiquitin in the asbcence of the immunoprecipitated protein (lane 1 in figures 3G and 3H, in the IP and the IB for Ub).B) These experiments are performed with overexpressed HA-Ub-K63, but the immunoblot is performed with anti-Ub. The immunoblot is not specific for HA-Ub-K63. In addition, these experiments do not show the covalently modified form of ubiquitinated RIG-I (or TRIM25 in figure 3H).C) All these experiments need to show the levels of overexpressed HA-Ub-K63 in the WCL to control for the transfection efficiencies.D) These experiments should also be performed without overexpression of ubiquitin to determine the endogenous ubiquitination of RIG-I

Since a new figure (new Figure 3) was incorporated into the main figures, the panel orders have changed as follow:

Original Figure 3 G → Figure 4H

Original Figure 3 H → Figure 4—figure supplement 1H

Original Figure S3 E → Figure 4—figure supplement 1I

New panel → Figure 4—figure supplement 1E

For comment A:

We have optimized our protocols and improved these images of polyubiquitination assays, in order to decrease the Ubs background in the ctrl lane. As shown in (Figure 4H, Figure 4—figure supplement 1E), the improved images clearly show that RTN3 inhibit RIG-I K63- linked polyubiquitination, although there is still a weak background, which might be caused by nonspecific interaction of the beads or anti-GFP antibody.

For comment B, C:

As the reviewer suggested, we detected the K63-linked or WT polyubiquitination with anti-HA antibody, and HA-K63 Ub bands of WCL also supplemented. Consistent results were obtained which again demonstrated that overexpressed RTN3 inhibits K63-linked polyubiquitination of RIG-I (Figure 4H, Figure 5B), and promotes K63-linked and WT polyubiquitination of TRIM25 (Figure 4—figure supplement 1H, 1I), upon VSV infection.

For the detection of “covalently modified form of ubiquitinated RIG-I (or TRIM25)”, we have optimized our protocols and improved the images to greatly diminish the nonspecific or degraded bands of ubiquitinated RIG-I. In fact, the background and nonspecific bands of Ub-linked RIG-I are closely related to the intensity in the western blot analysis of immunoprecipitated sample, if the exposure becomes weak, these background and nonspecific bands could not even be seen.

For comment D:

We have performed an additional experiment using anti-K63-linkage Specific Polyubiquitin Rabbit mAb (Cell Signaling Technology, #5621) to detect RTN3-mediated endogenous K63-linked polyubiquitination of RIG-I under VSV infection. A similar result was obtained (Figure 4—figure supplement 1E), clearly elucidating the inhibitory function of RTN3 toward endogenous RIG-I K63-linked polyubiquitination.

6) The conclusion that RTN3 is a "proresolving mediator' of immune and inflammatory responses, cannot be made when using only overexpression assays. The levels of RTN3 that are present in overexpression assays are not physiological and most likely do not represent real conditions. 'proresolving' mediator of inflammation is also an odd definition of these observation s. RIG-I has multiple negative regulators, which keep the pathway from overactivation. There is really no evidence that RTN3 acts at the 'resolving' stage of inflammation.

Thank you for your comments and suggestion. We agree that “proresolving mediator” is not an appropriate definition and then it has changed to “negative regulator”.

To strengthen our speculation of the inhibitory role of RTN3 in modulating RIG-I-mediated antiviral immune responses, we depleted RTN3 expression using shRNA in human PBMCs and challenged these cells with VSV. Consist with overexpression experiments, RTN3-depleted cells exhibited the enhanced phosphorylation level of TBK1, p65 and IRF3 upon VSV infection (Figure 3B), which leading the higher expression levels of type I interferon and proinflammatory cytokines (Figure 3C), eventually causing stronger resistance to viral infection (Figure 3D). In addition, we also provided data showing that RTN3 knockdown significantly enhanced K63-linked polyubiquitination of RIG-I under VSV infection (Figure 4—figure supplement 1F, 1G).

The following description has been added to the manuscript (Page 6, Lines 183-195):

“RTN3 depletion augments antiviral immune responses

To further elucidate the physiological function of RTN3 in antiviral responses, we depleted its expression by short hairpin RNAs (shRNAs) in human healthy donor PBMCs (Figure 3A), in which RTN3 normally undergoes upregulation during viral infection (Figure 1F). Within our experience, complete or vast obliteration of the expression of RTN3 by knockout or knockdown leads to cell death; hence, RTN3 expression were moderately depleted with transient shRNA transduction (Figure 3A). PBMCs were infected with control or RTN3 shRNA-encoding lentivirus for 12 h, and the infected cells were then settled for the following 24 h and subsequently infected with VSV-eGFP. Consistent with the decreased phosphorylation levels of TBK1, p65 and IRF3 in RTN3 overexpressing cells, RTN3 depletion significantly augmented the phosphorylation and activation of these critical kinases and transcription factors upon RNA viral infection (Figure 3B). Further RT-PCR analyses determined that the levels of IFNB1, IFIT2, IFIT1, TNF and the proinflammatory cytokines CCL20 and IL6 were enhanced by RTN3 knockdown during VSV infection (Figure 3C), while the amplification of VSV was compromised (Figure 3D). Collectively, depressed RTN3 expression facilitates antiviral innate immune and inflammatory responses upon viral infection.”

And (Page 7, Line 236-237):

“In contrast, RTN3 depletion enhanced the K63-linked polyubiquitination level of RIG-I (Figure 4—figure supplement 1F, 1G).”

7) The model shown in figure S7. This implies that there are two different cells, one that is undergoing 'inflammation' and another different cell that is 'resolving' inflammation. This also is not really supported by any data. The authors would need to do single-cell an analysis of infected vs non-infected cells, as well as more detailed analysis of which cytokines are controlling inflammation or dampening neutrophil numbers. The 'resolving' cell could have nothing to do with infection, and could be responding to cytokines. In addition, this conclusion would also need the use of knockout mice.

Thank you for your insightful comment and suggestion. We agree that our model that drawing two different cells to symbolize “inflammation” and “resolving inflammation” was imprecise, since the inflammation and resolution happen at the tissue level but not in a single cell. Our studies reveal that RTN3 negatively regulates RIG-I-mediated antiviral immune and inflammatory responses, thus the working model was reprogrammed and improved to diagram the inhibition and mechanism of RIG-I mediated antiviral responses by RTN3.

We moved it (the previous Figure S7) to main figures as Figure 8 and figure legend was added.

Finally, RTN3 knockout mice is not available currently, probably because RTN3 is an essential housekeeping gene for cell survival and RTN3 knockout causes cell death.

Following description had been added to the manuscript:

“Figure 8. The working model of the inhibition of RIG-I-mediated antiviral responses by RTN3

Upon RNA virus infection, the RIG-I signalosome is activated by TRIM25-mediated K63-linked polyubiquitination and then triggers IRF3 and NF-κB activation and the expression of interferons, cytokines and inflammatory factors. On the other hand, RTN3 is upregulated by acute viral infection and interferon/inflammatory factors in an ER stress-dependent manner. RTN3 is subsequently aggregated on the endoplasmic reticulum, interacts with RIG-I and TRIM25, rapidly diminishes the K63-linked polyubiquitination of RIG-I and hence suppresses antiviral and inflammatory responses. The inflammatory response is consequently mitigated, and the resolution of acute inflammation is initiated.”

Finally, we greatly appreciate all the reviewers and editors for the helpful comments and invaluable suggestions. We have improved our manuscript according to the comments and suggestions and addressed all of the points as described above.

References:

Grumati, P., G. Morozzi, S. Holper, M. Mari, M.I. Harwardt, R. Yan, S. Muller, F. Reggiori, M. Heilemann, and I. Dikic. 2017. Full length RTN3 regulates turnover of tubular endoplasmic reticulum via selective autophagy. *eLife* 6:

Kuniyoshi, K., O. Takeuchi, S. Pandey, T. Satoh, H. Iwasaki, S. Akira, and T. Kawai. 2014. Pivotal role of RNA-binding E3 ubiquitin ligase MEX3C in RIG-I-mediated antiviral innate immunity. *Proc Natl Acad Sci U S A* 111:5646-5651.

Yan, J., Q. Li, A.P. Mao, M.M. Hu, and H.B. Shu. 2014. TRIM4 modulates type I interferon induction and cellular antiviral response by targeting RIG-I for K63-linked ubiquitination. *J Mol Cell Biol* 6:154-163.